# Posterior parietal cortex oscillatory activity reflects persistent spatial memory impairments induced by early hippocampal amyloidosis in male mice

Souhail Djebari[1], Ana Contreras[1], Victor Castro-Andrés[2], Raquel Jimenez-Herrera[1], Guillermo Iborra-Lázaro[1], Raudel Sánchez-Campusano[2] (ID), Lydia Jiménez-Díaz[1] (ID) and Juan D. Navarro-López[1] (ID)

[1]*Neurophysiology & Behavior Lab, Instituto de Investigación Sanitaria de Castilla-La Mancha (IDISCAM) and Institute of Biomedicine (IB-UCLM), School of Medicine of Ciudad Real, University of Castilla-La Mancha, Ciudad Real, Spain*
[2]*Department of Physiology, Anatomy and Cell Biology, Pablo de Olavide University, Seville, Spain*

Handling Editors: Nathan Schoppa & Valentina Mosienko

The peer review history is available in the Supporting information section of this article (https://doi.org/10.1113/JP286196#support-information-section).

**Abstract figure legend** Schematic representation of the spatiotemporal progression of the deleterious effects of early amyloidosis. Beginning as early as 1 h after i.c.v. injection of oligomeric $A\beta_{1-42}$, alterations in hippocampal function emerge, including aberrant oscillatory activity, deficits in synaptic plasticity and impairments in habituation memory. By 3 days post-injection, these pathological changes extend to the posterior parietal cortex, where atypical patterns of oscillatory activity persist for at least 12 days post-injection. These sustained alterations in cortical activity may contribute to the observed long-lasting spatial memory deficits.

**Abstract** In early stages of Alzheimer's disease (AD), soluble amyloid-$\beta$ (A$\beta$) is a key player disrupting neuronal activity and contributing to cognitive decline in advanced stages of the disease. Although the hippocampus has been a central focus in prior research because of its susceptibility

S. Djebari, A. Contreras and V. Castro-Andrés contributed equally to this work and share first authorship.

R. Sánchez-Campusano, L. Jiménez-Díaz and J. D. Navarro-López contributed equally to this work and share last authorship.

This article was first published as a preprint. Djebari S, Contreras A, Jimenez-Herrera R, Castro-Andres V, Iborra-Lazaro G, Sanchez-Campusano R, Jimenez-Diaz L, Navarro-Lopez JD. 2023. Posterior parietal cortex oscillatory activity shapes persistent spatial memory impairments induced by soluble amyloid-$\beta$ oligomers. Research Square. https://doi.org/10.21203/rs.3.rs-3791891/v1

The Journal of Physiology

to A$\beta$-induced alterations, a comprehensive understanding of the temporal progression of early AD pathology requires exploring interconnected brain regions. The posterior parietal cortex (PPC), collaborating closely with the hippocampus and involved in various memory processes, particularly spatial memory formation, holds particular significance. Investigating the function of the PPC is imperative because it may contribute to early AD characteristics and provide a more holistic perspective on disease progression. To address this gap, we examined the relationship between neural oscillations and memory processes in both the PPC and hippocampus, in a mouse model of early hippocampal amyloidosis generated by intracerebroventricular oligomeric A$\beta_{1\text{-}42}$ (oA$\beta_{1\text{-}42}$) injection. By performing *in vivo* oscillatory activity recordings from these regions in alert animals, together with spatial and habituation memory tests (Barnes maze and open field habituation), we found oA$\beta_{1\text{-}42}$ to induce significant alterations in PPC oscillatory activity. These changes emerged several days after hippocampal disturbances showed as aberrant synaptic plasticity and network activity. Additionally, significant alterations of stereotyped behaviours were not found. Our results provide an electrophysiological substrate for persistent spatial memory deficits and the temporal progression pattern of the early deleterious effects caused by A$\beta$. Furthermore, investigating PPC oscillatory activity might be a valuable approach for early detection and intervention in AD.

(Received 25 June 2024; accepted after revision 9 February 2026; first published online 8 March 2026)

**Corresponding authors** R. Sánchez-Campusano: Department of Physiology, Anatomy and Cell Biology, Pablo de Olavide University, Seville, Spain.    Email: rsancam@upo.es

L. Jiménez-Díaz and J. D. Navarro-López: Neurophysiology & Behaviour Lab, Instituto de Investigación Sanitaria de Castilla-La Mancha (IDISCAM) and Institute of Biomedicine (IB-UCLM), School of Medicine of Ciudad Real, University of Castilla-La Mancha, Ciudad Real, Spain.    Email: lydia.jimenez@uclm.es, juan.navarro@uclm.es

**Key points**

- Posterior parietal cortex (PPC), in close collaboration with the hippocampus, has been implicated in various memory processes disrupted in early Alzheimer's disease models.
- A mouse model of early Alzheimer's-like hippocampal amyloidosis generated by intra-cerebroventricular oligomeric A$\beta_{1\text{-}42}$ (oA$\beta$1 42) injection was used to examine the relationship between neural oscillations and memory processes in both the PPC and hippocampus.
- oA$\beta_{1\text{-}42}$ induces alterations in spatial and habituation memory, associated with PPC aberrant oscillatory activity, several days after hippocampal synaptic plasticity and network activity disturbances were found.
- We provide an electrophysiological PPC-mediated substrate for persistent spatial memory deficits and the temporal progression pattern of the early oscillatory deleterious effects caused by A$\beta$.

## Introduction

The posterior parietal cortex (PPC) is an associative cortical area situated in the posterior segment of the parietal lobe. It functions as a hub for processing sensory inputs, particularly from somatic and visual regions, synthesising this information to construct a representation of the body's position in space (Buneo

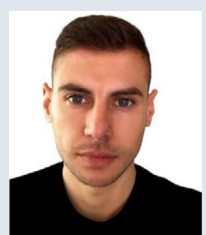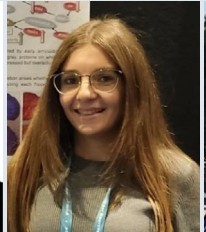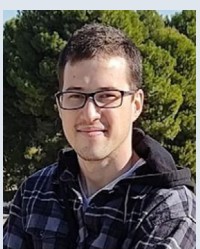

**Souhail Djebari** earned his PhD in Health Sciences from the University of Castilla-La Mancha. He is currently an associate professor at the Faculty of Medicine in Ciudad Real, where his research focuses on the early stages of Alzheimer's disease. **Ana Contreras** received her PhD in Neurosciences from CEU-San Pablo University. As a postdoc, her research focuses on the early stages of Alzheimer's disease and the discovery of potential biomarkers for this disease. **Víctor de Castro** is completing his PhD at Pablo de Olavide University, specialising in neuroscience. His research focuses on neural data science, neurophysiological modelling and computational neuroscience.

& Andersen, 2006; Jacobs et al., 2012). Traditionally, the PPC has been liked to various cognitive functions, encompassing visual perception, spatial attention and visual guidance (Lefco et al., 2020). This area can be further dissected into two primary subregions: the dorsal PPC (dPPC), implicated in top–down attention driven by internal goals and the ventral PPC (vPPC), mediating bottom–up, stimulus-driven attention (Ciaramelli et al., 2020; Yang et al., 2022).

Recent studies have unveiled the role of the PPC in episodic memory encoding and retrieval, in both humans and rodents (Brodt et al., 2016; Cabeza, 2008; Cabeza et al., 2008; Lefco et al., 2020). Traditionally, these functions have been ascribed to the hippocampus (Moser & Moser, 1998), but evidence from patients with parietal lesions began to suggest a possible involvement of the parietal cortex in this type of memory (Cabeza et al., 2008). In this sense, successful episodic memory formation and recall require a delicate interplay between the emergence of stored memory traces, overseen by the hippocampus, and the top–down control based on retrieval goals, attributed to the prefrontal cortex (Preston & Eichenbaum, 2013). Additionally, the PPC appears to play a pivotal role in episodic memory retrieval and updating, influencing new memories based on current experiences (Myskiw & Izquierdo, 2012; Rogers & Kesner, 2006; Suzuki et al., 2022). Within this framework, spatial memory acquires significant importance. It has been observed that, although the hippocampus oversees the initial processing phases of spatial memory, the PPC is essential for its long-term retention. This indicates that information transfer between these regions is necessary for accurate processing, highlighting the integrated role of these brain areas in memory function (Kesner, 2009).

The investigation into the neural basis of these memory processes directs attention to neuronal oscillations, specifically theta ($\theta$) and gamma ($\gamma$) rhythms, which are fundamental to memory modulation (Buzsáki & Moser, 2013; Colgin & Moser, 2010; Nyhus & Curran, 2010). Consequently, abnormal oscillatory activity could contribute to the pathophysiology of memory-related neurodegenerative diseases (Hector et al., 2023; Palop & Mucke, 2016).

Thus, understanding the interplay between PPC and hippocampal oscillatory activity and memory has significant implications for neurodegenerative diseases such as Alzheimer's disease (AD) (Giannakopoulos et al., 2000). AD is a progressive neurodegenerative disorder characterised by the accumulation of amyloid-$\beta$ (A$\beta$) senile plaques and neurofibrillary tangles in the brain, ultimately leading to neuronal dysfunction and cell death (Jeremic et al., 2021). Memory impairment is a hallmark symptom of AD, with the hippocampus being one of the earliest affected areas (Vyas et al., 2020). Previous investigations focused on unravelling the initial

phases of AD have highlighted that soluble A$\beta$ species detrimentally affect hippocampal oscillatory activity, contributing to cognitive deficits alongside alterations in synaptic plasticity (Sánchez-Rodríguez et al., 2017). In the context of the PPC, investigations have shown that patients with mild cognitive impairment (MCI), a pre-AD stage, exhibit grey matter loss in this area compared to healthy subjects (Jacobs et al., 2012). Additionally, significantly reduced connectivity between the PPC and the hippocampus has been observed in both MCI and early stages of AD (Ilardi et al., 2022; La Joie et al., 2014). Consequently, dysfunction within the PPC-hippocampus network may serve as an early marker of AD pathology.

Nonetheless, the potential alterations in cortical and hippocampal oscillatory activity during the early stages of AD and its subsequent progression over time remain largely unexplored. Therefore, the present study aimed to examine how early acute amyloidosis longitudinally impacts the oscillatory activity of both the hippocampus and the PPC, as well as to assess the role played by these two regions in shaping memory and learning abilities.

## Methods

### Ethical approval

All experimental procedures were reviewed and approved by the Ethical Committee for Use of Laboratory Animals of the University of Castilla-La Mancha (PR-2023-25, PR-2021-12-21 and PR-2018-05-11) and conducted in accordance with European Union guidelines (2010/63/EU) and Spanish regulations governing the use of laboratory animals in chronic experiments (RD 53/2013 on the care of experimental animals: BOE 08/02/2013 and updated on 24/02/2021). All efforts were made to minimise animal pain and suffering. All researchers involved in this study were fully aware of the ethical principles under which *The Journal of Physiology* operates and complied with the animal ethics checklist established by *The Journal of Physiology*, the International Council for Laboratory Animal Science and the ARRIVE guidelines.

### Animals

Male C57BL/6 adult mice ($n = 58$; 2–5 months old; 20–35 g) were used (RRID:MGI:5656552; Charles River, Wilmington, MA, USA). Animals were maintained under a 12:12 h light/dark photocycle with *ad libitum* access to food and water. The temperature was controlled at 21 ± 1°C, and humidity was maintained at 50% ± 7%. Prior to surgery, mice were housed together (up to five per cage), but, after surgery, they were individually housed to ensure their well-being. All experimental procedures were conducted at consistent time intervals to mini-

mise potential circadian rhythm influences. Mice were regularly handled to reduce stress during experimental procedures.

## Surgical procedure to generate an early amyloidosis model

All animals underwent surgery to enable chronic drug delivery. Among these, some were also implanted with electrodes for *in vivo* recordings (cohorts 1 and 2A) (Fig. 1*B*). The surgical procedure followed the methodology previously described (Jiménez-Herrera et al., 2023; Sánchez-Rodríguez et al., 2017). Initially, mice were anesthetised using a calibrated R580S vaporiser (RWD Life Science, Solana Beach, CA, USA) with a flow rate: $0.5$ L min$^{-1}$ O$_2$. Anaesthesia induction employed 4% isoflurane (#13400264, ISOFLO; Proyma S.L., Madrid, Spain), which was then maintained at 1.5% isoflurane throughout the surgery. Following surgery, mice received I.M. buprenorphine ($0.01$ mg kg$^{-1}$; #062009; Buprenodale, Albet, Spain) for pain management and a topical healing cream (Blastoestimulina; Almirall, Baecelona, Spain).

For I.C.V. administration of A$\beta$, animals underwent the implantation of a stainless steel, 26-G guide cannula (Plastics One, Roanoke, VA, USA) into the right ventricle (1 mm lateral and 0.5 mm posterior to bregma; depth from brain surface, 1.8 mm) (Paxinos & Franklin, 2004). The final positioning of the cannula was confirmed using Nissl staining (Fig. 1*A*).

For chronic recording, bipolar electrodes, constructed from 50 μm Teflon-coated tungsten wire (Advent Research Materials, Eynsham, UK), were surgically implanted in each animal (Fig. 1*A*). Two separate cohorts of animals were utilised, each with electrodes targeting distinct locations. In the first cohort, a subdural recording electrode was directed towards the surface of dorsomedial PPC (1.2 mm lateral and 1.8 mm posterior to bregma). In the second cohort (cohort 2A), the recording electrode was aimed at the stratum radiatum beneath the CA1 area (1.2 mm lateral and 2.2 mm posterior to bregma; depth from brain surface, 1.0–1.5 mm) (Paxinos & Franklin, 2004). In all cases, a 0.1 mm silver wire was securely attached to the skull as ground, and both electrodes and grounding were then affixed to a 4-pin socket (Mouser Electronics, Mansfield, TX, USA), which was further attached to the skull using dental cement.

Following the surgical procedures, mice were allowed a recovery period of at least 1 week before any experimental interventions. Subsequently, animals were randomly assigned to either the A$\beta_{1-42}$ experimental group or the control group receiving vehicle [phosphate-buffered saline (PBS)]. A$\beta_{1-42}$ oligomers [oligomeric A$\beta_{1-42}$ (oA$\beta_{1-42}$)] were prepared in PBS as described elsewhere (Jiménez-Herrera et al., 2023; Pavliukeviciene et al., 2019;

Vosough & Barth, 2021) and procured from Abcam (Cambridge, UK) (#ab120301). To administer oA$\beta_{1-42}$, alert mice received a 3 μL I.C.V. injection of the drug at a concentration of 1 μg/μl. The dose was selected based on previous studies that demonstrated its effectiveness and safety (Jiménez-Herrera et al., 2023; Sánchez-Rodríguez et al., 2017, 2019, 2020). The injection was performed through an insertion cannula within the implanted guide cannula using a Hamilton syringe, at a rate of $0.5$ μL min$^{-1}$. Verification of the correct administration of oA$\beta_{1-42}$ was confirmed through immunohistochemistry staining using a mouse anti-A$\beta_{1-42}$ antibody (dilution 1:500; #803001; Biolengend, San Diego, CA, USA) as shown in Fig. 1*A*. For this procedure, animals were deeply anaesthetised with halothane (fluothane; AstraZeneca, Cambridge, UK) before decapitation. The injection of oA$\beta_{1-42}$ was performed the day before training (i.e. day 0) of the corresponding behavioural task to assess the full spectrum of memory processing.

## *In vivo* electrophysiological recordings and analysis

Local field potential (LFP) activity was recorded using Dagan Corporation EX4-400 Quad Differential amplifiers (Dagan Corporation, Minneapolis MN, USA) at a bandwidth of 0.1–10 kHz, through a high-impedance probe ($2 \times 1012$ Ω, 10 pF) in alert behaving mice placed in a plywood box ($35 \times \times 20$ cm) and in the absence of any electrical stimulation. To ensure that the animals' natural behaviour and the accuracy of behavioural test measurements were not disturbed, recordings were conducted immediately following the behavioural tests each day. These recordings were digitally stored on a computer via an 8-channel analogue-to-digital converter (CED 1401-plus; CED, Cambridge, UK).

In the first cohort of animals, LFPs were recorded from the PPC for 10 min before the I.C.V. injections (pre-treatment values) and at 1, 3 and 12 days post-injection, immediately after the behavioural test each day, with the animals awake and freely moving (Fig. 1*C*). These time points were selected for their relevance in the Barnes maze paradigm: at the beginning and end of the training phase (days 1 and 3, respectively) and at the test phase (day 12). From these recordings, 5 min long segments that were free from any artefacts were chosen for detailed analysis. Only those animals ($n = 10$) that provided a clear and uninterrupted 5 min segment of recording were considered for further analysis.

To analyse the LFP recordings, computer programs (Spike2 and SIGAVG; CED) were used to display the LFP data in the time domain and export it in ASCII format. The LFP recordings were sampled at 5 kHz, and spectral analyses were conducted in the following frequency bands: delta ($\delta$; 1–4 Hz), $\theta$ (4–12 Hz), beta ($\beta$; 12–30 Hz),

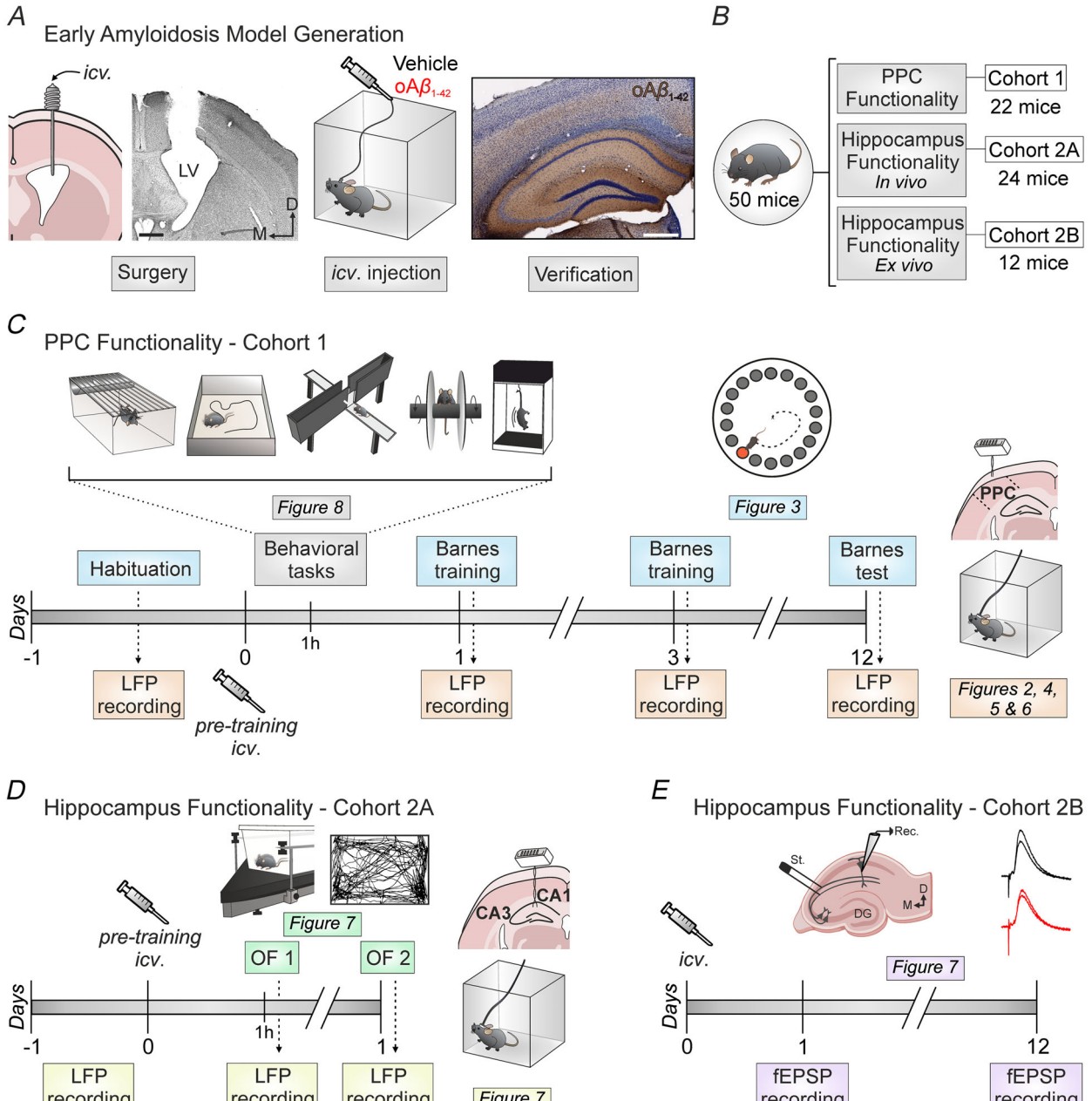

**Figure 1. Experimental design**

*A*, to generate the early amyloidosis model, a stainless-steel guide cannula was surgically implanted for *in vivo* I.C.V. drug administration (vehicle or oA$\beta_{1-42}$) into the left ventricle. The cannula's placement was histologically verified (left), and oA$\beta_{1-42}$ immunochemistry staining confirmed the successful induction of amyloidosis (right). Scale bars = 500 µm. *B*, schematic representation of the number of animals assigned to each cohort. *C*, timeline of the first cohort of animals that underwent a series of experiments after I.C.V. injection of the corresponding treatment to assess PPC functionality: assessments of motor and anxiety-like behaviour (day 0, starting 1 h post-I.C.V.; in order of performance: spontaneous behaviour, open field locomotion, elevated plus maze, rotarod performance test and tail suspension test), Barnes maze task (days 1, 3 and 12 post-I.C.V. following behavioural tasks), and *in vivo* oscillatory activity recording in the PPC with surgically implanted electrodes (days −1 pre-I.C.V. and 1, 3 and 12 post-I.C.V.). *D* and *E*, timeline of the second cohort of animals that underwent a series of experiments after I.C.V. injection of the corresponding treatment to assess hippocampus functionality. *D*, cohort 2A: open field habituation test (1 h and 1 day post-I.C.V. following behavioural tasks) and *in vivo* oscillatory recording in the hippocampus with surgically implanted electrodes (days −1 pre-I.C.V. and 1 h and 1 day post-I.C.V.). *E*, cohort 2B: *ex vivo* assessment of hippocampal LTP (days 1 and 12 post-I.C.V.). D, dorsal; DG, dentate gyrus; I.C.V., intracerebroventricular; LV, lateral ventricle; LTP, long-term potentiation; M, medial, oA$\beta_{1-42}$, amyloid-$\beta_{1-42}$ oligomers; PPC, posterior parietal cortex; Rec, recording; St, stimulation.

low $\gamma$ (low $\gamma$; 30–48 Hz) and high $\gamma$ (high $\gamma$; 52–150 Hz). LFP recordings processing involved frequency domain analysis of the power spectra using fast Fourier transforms (FFTs) and time-frequency domain analysis of the power spectrograms using multitapered Fourier transforms (mTFTs). The two spectral analyses (FFT and mTFT) were performed under the four experimental conditions (days: PRE, D-1, D-3 and D-12) and in both groups of mice (oA$\beta_{1-42}$ and vehicle; $n = 10$ mice per group). For FFT analysis, 10 s epochs ($N_E = 18$ trials) covering a 3 min analysis window of LFP recordings acquired from 10 mice per group (oA$\beta_{1-42}$ and vehicle) across the 4 days were used, allowing power spectra to be plotted and power values to be calculated. By means of FFT, it was possible to obtain both the raw spectra and the normalised power spectra with respect to the total power. By normalising, the effect of absolute power differences between the groups is removed, allowing the comparison to focus on the relative distribution of power across different frequencies.

For mTFT analysis of each time-frequency spectrogram, a full 5 min segment of LFP ($N_E = 6$ epochs of 300 s each) was used across the four experimental conditions (days: PRE, D-1, D-3 and D-12) and both groups of mice (oA$\beta_{1-42}$ and vehicle). The mTFT method uses a family of Slepian sequences called tapers. These sequences have the property that, for a given short time-window of data ($T = 2$ s) and a narrow frequency-bandwidth ($W = 1$ Hz), the first $K$ tapers ($K = 2TW – 1$, number of tapers $K = 3$) are optimally concentrated in the frequency range $[f – W, f + W]$ and the multitapers method provides estimates with good bias-variance characteristics. Therefore, because Slepian sequences are mutually orthogonal and the trials are interchangeable, the spectrum estimates computed from the different tapers and trials are statistically independent and averaging over them reduces the variance of the estimate. Thus, multitaper estimates of the spectrogram involving $K$ tapers and $N_E$ epochs are based on computing $m = N_E \times K$ Fourier transforms (FTs).

Next, an analysis of the spectra of the LFP recordings from the PPC was performed. Figure 2*A–C* displays the mean power spectra of the LFPs recorded from PPC of mice in the two groups (vehicle and oA$\beta_{1-42}$) across the different time points (PRE, D-1, D-3 and D-12). Figure 2*D–L* shows log-log plots of the aperiodic components of the LFP spectra are shown for the two range of frequencies (range 1: 1–48 Hz; range 2: 52–150 Hz). Note that power-frequency relationship $P(f) = 1/f$ (or law of slope equal to $-1$ in the log-log plot) does not hold in the frequency range of 1–48 Hz (slopes in the range $[-0.56, -0.33]$; mean $= -0.46$). On average, the point cloud was best fitted to a power-frequency relationship of the type $P(f) = C/\sqrt{f}$

by slope approximation ($\alpha \approx -0.5$). These findings indicate that the aperiodic components of the spectra are not suitable for characterising the spectral features of the recorded LFPs. Therefore, in the present study, we focused our analysis on the periodic components, particularly the fundamental harmonic within the $\theta$ (4–12 Hz) frequency band.

Script packages were developed by one of the investigators (RS-C) using MATLAB, version 9.12, R2022a (The MathWorks, Natick, MA, USA). Customised scripts from Chronux, versions 2.11/R2014 and 2.12/R2018(http://chronux.org/) (Bokil et al., 2010)] were also employed for systematic spectral analysis (Fernández-Lamo et al., 2016; Jurado-Parras et al., 2013). Probability maps for comparing pairs of spectrograms were generated as described in previous studies (Conde-Moro et al., 2019; Fernández-Lamo et al., 2016; Lintas et al., 2021; Reus-Garcia et al., 2021). Pie chart representations of probability density [for types $\pm1$ ($P < 0.05$) and 0 ($P > 0.05$) inferences] for selected frequency bands were obtained based on Jackknifed estimation of variance between pairs of spectrograms (time-frequency plots of the LFP powers), covering all frequency values (resolution, $dF = 0.5$ Hz) within each time window (resolution, $dt = 0.5$ s). Additionally, a powerful band-stop IIR filter (notch filter) with cutoff frequencies at 48 and 52 Hz was implemented to eliminate 50 Hz interference noise.

In a second cohort of animals (cohort 2A), LFPs were recorded from the hippocampal CA1 area for 10 min before (pre-treatment values) and 1 h and 1 day post-i.c.v. treatment (Fig. 1*D*). Up to 5 min of artefact-free recordings were selected for subsequent spectral analysis using Spike2 (CED). The FFT was also used to transform the signal into frequency-dependent functions. All LFP spectral powers obtained by means of FFT were normalised using values from the pre-treatment day as 100%.

## Barnes maze task

The Barnes maze (LE851BSW; Panlab, Barcelona, Spain) was employed to assess short- and long-term spatial memory in the first cohort of animals (Fig. 1*C*), as previously described (Jiménez-Herrera et al., 2023). This test was chosen because of its effectiveness in spatial, hippocampal-dependent memory evaluation and its suitability for assessing long-term memory performance (Gawel et al., 2019).

In brief, each trial commenced with the mouse positioned at the centre of a maze measuring 92 cm in diameter and located 1 m above the floor. The maze featured 20 escape holes, each measuring 5 cm in diameter, distributed along its periphery. Spatial cues, consisting of

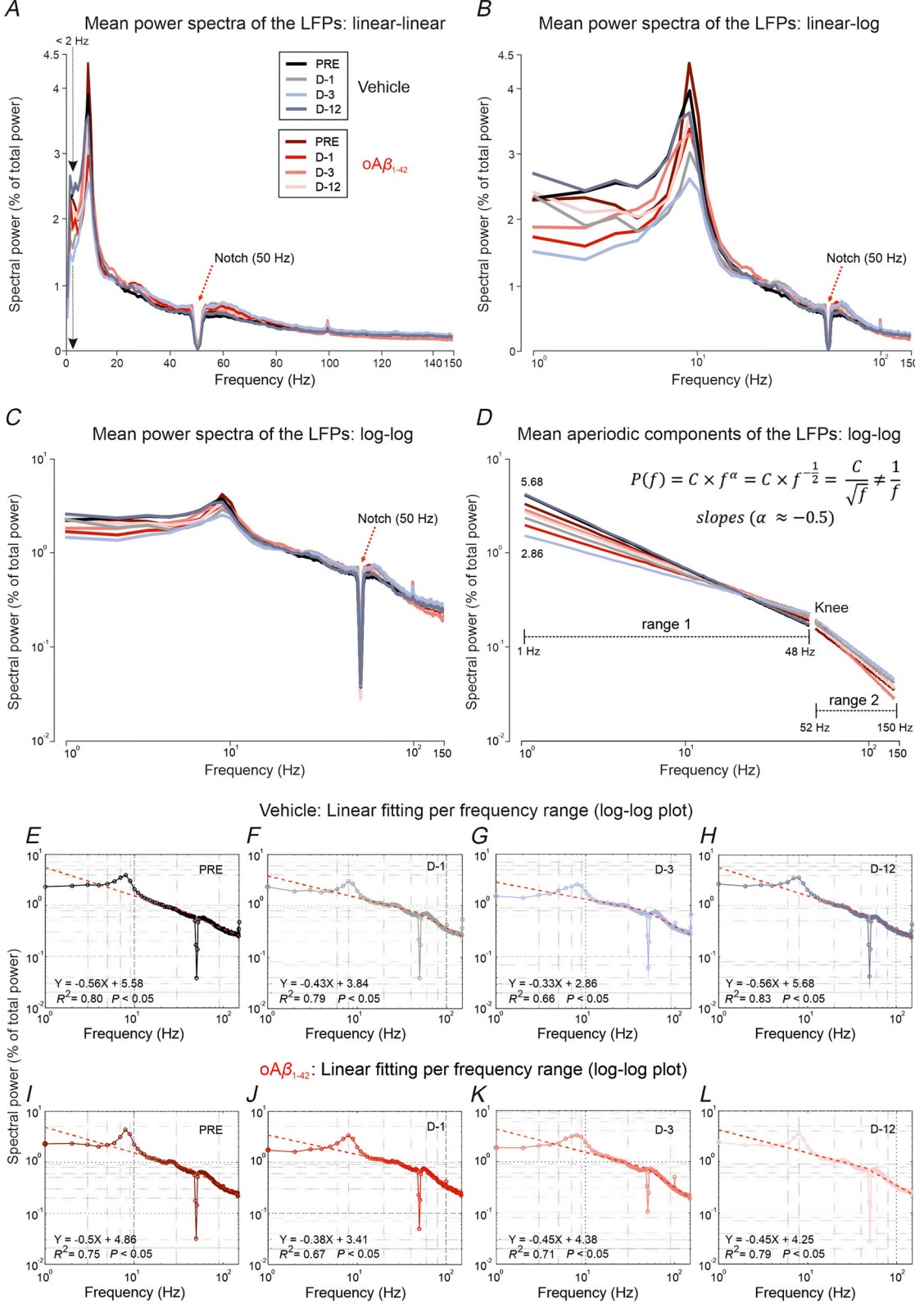

**Figure 2. Different representations of the mean power spectra of the LFPs recorded from PPC of the mice ($n = 10$ per group) in the two groups (vehicle and oA$\beta_{1\text{-}42}$) across the days (PRE, D-1, D-3 and D-12)**
*A*, linear–linear plot of the spectra showing the spectral component in the range 0–2 Hz with peak at 1 Hz (see vertical black arrows, <2 Hz) within the delta ($\delta$; 0–4 Hz) band. *B*, linear–log plot of the spectra where the *x*-axis starts at 1 Hz because the base-10 logarithmic function of power is undefined for small values (<1 Hz) of frequency.

This is the representation adopted for this study because it focuses on the fundamental periodic component (~8 Hz) in the theta ($\theta$; 4–12 Hz) frequency band. *C*, log–log plot of the spectra with low dispersion (intercepts in the range [2.86, 5.68]) that makes it difficult to differentiate the spectra. *D*, log–log plot of the aperiodic components of the LFP spectra for the two range of frequencies (range 1: 1–48 Hz; range 2: 52–150 Hz). Note that power–frequency relation $P(f) = 1/f$ (or law of slope equal to $-1$ in the log–log plot) does not hold in the frequency range of 1–48 Hz (slopes in the range [$-0.56$, $-0.33$]; mean = $-0.46$). On average, the point cloud was best fitted to a power–frequency relationship of the type $P(f) = C/\sqrt{f}$ by slope approximation ($\alpha \approx -0.5$). *E-L*, each panel corresponds to the linear fit between power *vs.* frequency points in each experimental condition and session. Linear equations ($Y = \text{slope}*X + \text{Intercept}$) and the corresponding determination coefficients ($R^2$) are indicated. All representations correspond to power spectra normalised with respect to the total power (see *y*-axis). Plots in (*A*) to (*C*) show the notch (see red arrows) resulting from filtering the 50 Hz interference noise, with cutoff frequencies at 48 and 52 Hz. LFP, local field potential; oA$\beta_{1-42}$, amyloid-$\beta_{1-42}$ oligomers; PPC, posterior parietal cortex.

circles and squares of distinct colours, were positioned around the room. To prevent odour cues from influencing the trials, the maze was cleaned with 70% ethanol between each session.

The day prior to I.C.V. injection, a habituation phase was conducted, allowing the mice to explore the maze for 90 s. Subsequently, three training days were implemented, commencing on the day following I.C.V. treatment, with each day consisting of three trials separated by 15 min intervals. During each trial, the mice had 3 min to explore the maze or until they located the escape hole (measured by latency and probability of reaching the escape hole), which contained a box measuring 17.5 × 7.5 × 8 cm. Finally, on day 12 post-treatment, a single 90 s trial was conducted as a memory test. In this session, no escape holes were available, and the latency and probability of reaching the target hole were recorded. All sessions were recorded and analysed using the Barnes-Smart video tracking software (Panlab).

### Open field habituation task

The open field (OF) habituation task was employed to evaluate a non-associative hippocampal-dependent learning process, specifically the exploratory habituation to a novel environment (Leussis & Bolivar, 2006), in a subset of animals from the second cohort (cohort 2A) (Fig. 1*D*). This particular test was selected due to its capability to facilitate both intra- and intersession assessments, comprehensively addressing different stages of memory processing that significantly involve the hippocampus (Daenen et al., 2001; Wright et al., 2004).

Briefly, 1 h after the I.C.V. injection, a training session (OF1) was conducted. During this session, mice were placed in the centre of a square acrylic box (measuring 38 × 22.5 × 4 cm at the plexiglas base arena and 43.5 × 27.5 × 22.5 cm at the top) and allowed to freely explore the environment for a duration of 15 min. On the following day, a retention phase (OF2) was carried out, where the animals were reintroduced to the same environment for an equal amount of time.

To measure the change in exploratory behaviour between these sessions, a Laboratory Animal Behaviour Observation Registration and Analysis System (LABORAS) apparatus (LABORAS; Metris, Hoofddorp, The Netherlands) was utilised. This system transforms the mechanical vibrations produced by the animals' movements into electrical signals through a sensing platform located beneath the cage.

### Spontaneous behaviours

To evaluate the overall health state of the animals, the LABORAS was also utilised for stereotyped and locomotion behavioural testing (Fig. 1*C*). One hour after the I.C.V. injection, mice from cohort 1 were placed in a rectangular LABORAS cage (measuring 23.5 × 17.5 × 4 cm at the plexiglas base arena and 26.5 × 21 × 10 cm at the top) for a single 15 min trial, as previously described (Djebari et al., 2021).

Motor activity was quantified through measurements of locomotion, climbing and rearing, whereas grooming behaviour was used as an indicator of stress. All data were digitised and analysed using the LABORAS software (Metris).

### OF locomotion

The LABORAS system was also used to evaluate additional parameters indicative of the general health state in animals from cohort 1. Right after evaluating spontaneous behaviours, mice were placed in a rectangular cage box (measuring 38 × 22.5 × 4 cm at the plexiglas base arena and 43.5 × 27.5 × 22.5 cm at the top) for a 15 min session. The distance travelled, serving as an indicator of motor activity, and the time spent in the periphery of the box (defined as a perimeter of 5.4 cm from the walls), used to evaluate anxious behaviour, were analysed (Fig. 1*C*).

### Elevated plus maze

Subsequent to the OF locomotion assessment, the elevated plus maze (LE 842; Panlab) was utilised to evaluate anxiety-like behaviours (La-Vu et al., 2020) (Fig. 1*C*). Mice

from cohort 1 were placed in the centre of the maze, facing one of the open arms of a cross-shaped methacrylate platform with two open arms (65 × 6 cm; without walls) and two enclosed arms (65 × 6 cm; 15 cm high opaque walls) raised 40 cm above the floor. During a single 5 min session, the number of entries into the open arms and the time spent there were calculated to measure anxiety-like behaviour. Additionally, the number of entries into the closed arms and the total entries (open + closed arms) were recorded as measures of locomotor activity (Lopes et al., 2021). All data were digitised and analysed using SMART, version 3.0 (Panlab).

### Rotarod performance test

To assess the impact of the treatment on co-ordination and motor function, the rotarod apparatus (LE 8500; Panlab) was employed (Fig. 1*C*). The day before treatment administration, mice from cohort 1 were introduced to a 3 cm diameter black striated rod positioned 20 cm above the floor, which rotated at a constant low-speed of 6 r.p.m. for 1 min, as a habituation exercise. Following the i.c.v. injections and elevated plus maze test, the mice were tested for their ability to remain on the rod across six consecutive trials, with the rod accelerating from 4 to 40 r.p.m. over a 2 min period in each trial. Data were digitised and analysed using Sedacom software (Panlab).

### Tail suspension test

The tail suspension test was performed to assess depression-like behaviours (Fig. 1*C*) (Cryan et al., 2005), immediately following the rotarod performance evaluation in mice from cohort 1. Animals were suspended by their tails 19 cm above the ground in the tail suspension apparatus (BIO-TST5; Bioseb, Pinellas Park, FL USA), within a polyvinyl chloride chamber (50 × 15 × 30 cm) for 6 min. Immobility time was recorded by the strain sensor as a measure of depression-like behaviour, along with assessments of energy and power of movement to evaluate motor function. All data were digitised and analysed using the BIO-TST 4.0 software (Bioseb).

### *Ex vivo* field EPSP (fEPSP) recordings

In a subset of animals from the second cohort (cohort 2B), staggered injections were performed over several days to explore CA3–CA1 synapse plasticity, either 1 day (short-term) or 12 days (long-term) after i.c.v. injections. Coronal hippocampal slices were then prepared as previously described (Djebari et al., 2021). These particular animals were excluded from further tests to avoid potential interferences and to preserve the integrity of the hippocampal slices for electrophysiological analysis, particularly concerning the impact of *in vivo* electrode implantation. After deep anaesthesia with halothane (fluothane; AstraZeneca) and decapitation, brain was quickly removed and submerged in oxygenated (95% $O_2$–5% $CO_2$) ice-cold artificial cerebrospinal fluid (aCSF) containing (in mmol $L^{-1}$; all from Sigma, St Louis, MO, USA): 118 NaCl (#S9888), 3 KCl (#P3911), 1.5 $CaCl_2$ (#499 609), 1 $MgCl_2$ (#208 337), 25 $NaHCO_3$ (#S6014), 30 glucose (#G8270) and 1 $NaH_2PO_4$ (#S8282). The brain was placed on the stage of a vibratome (7000smz-2; Campden Instruments, Loughborough, UK) for the preparation of coronal slices (350 μm thick) containing the dorsal hippocampus. Subsequently, these slices were incubated for at least 1 h at room temperature (22°C) in aCSF.

For electrophysiological recordings, a single slice was transferred to an interface recording chamber (Warner Instruments, Holliston, MA, USA) that was continually perfused with oxygenated aCSF at a flow rate of 1 ml $min^{-1}$, using a peristaltic pump (Minipuls 3 Peristaltic Pump; Gilson, Middleton, WI, USA). After a stabilisation period of at least 10 min within the recording chamber, a borosilicate glass micropipette (1–3 MΩ; World Precision Instruments, Sarasota, FL, USA) connected to an extracellular recording amplifier (Neurolog system; Digitimer, Fort Lauderdale, FL, USA) was positioned in the stratum radiatum of CA1. Simultaneously, a tungsten concentric bipolar stimulating electrode (World Precision Instruments), connected to a Master-9 stimulator (AMPI, Jerusalem, Israel), was targeted at the Schaffer collateral pathway. Using a stimulus isolation unit (ISO/Flex; AMPI), biphasic, 60 μs long, square-wave pulses were employed as stimuli, adjusted to ∼35% of the intensity necessary to evoke the largest fEPSP response.

The baseline was established by collecting fEPSPs for 15 min. Long-term potentiation (LTP) was then induced using a high-frequency stimulation (HFS) protocol, which consisted of five, 1 s long, 100 Hz trains delivered at 30 s intertrain intervals. After HFS, fEPSPs were recorded for 60 min to evaluate the induction of LTP (Djebari et al., 2021). To align the effects of o$A\beta_{1-42}$ on long-term potentiation with the effects observed in the behavioural tasks, electrophysiology was conducted at 1 day (short-term) and 12 days (long-term) post-i.c.v. injection (Fig. 1*E*).

The data were displayed using an oscilloscope (MDO3000; Tektronix, Beaverton, OR, USA) and digitised using Spike2 (CED). Because synaptic responses were not affected by population spikes, the amplitude (i.e. the peak-to-peak value in mV during the rise-time period) of successively evoked fEPSPs was analysed using Signal software (CED).

## Statistical analysis

The data were expressed as the mean $\pm$ SD and subjected to statistical analysis using one- or two-way repeated measures ANOVA, as appropriate. Time and treatment served as within- and between-subjects factors, respectively, and Sidak's *post hoc* analysis was employed for multiple comparisons. Sphericity was tested (Mauchly's test) and was not violated in any case; thus, Greenhouse–Geisser corrections were not applied. For analysis involving only two independent groups, unpaired Student's *t* test was applied. In cases where data did not follow a normal distribution, the Mann–Whitney *U* test was used instead. Normal distribution of the data was checked with the Kolmogorov–Smirnov test, and homogeneity of variances was tested with Levene's test. For the Barnes maze task, the cumulative incidence of latency to find the escape hole was analysed using a Cox proportional hazard model, with group as a categorical covariate and day as continuous covariate. Additional statistical information related to Figs 4 and 5 is provided in the Results section.

$P < 0.05$ was considered statistically significant. All analyses were carried out using SPSS, version 24 (RRID:SCR_002865; IBM Corp., Armonk, NY, USA), Prism, version 8.3.1 (RRID:SCR_002798; Dotmatics, Boston, MA, USA) and the Statistics MATLAB Toolbox, version 9.12, R2022a (The MathWorks). Figures were generated using CorelDraw X8 Software (RRID:SCR_01 4235; Corel Corporation, Ottawa, ON, Canada).

# Results

## Disruption of cortical oscillatory activity by oA$\beta_{1-42}$ 3 days after I.C.V. administration

The co-ordinated activity between the PPC and the hippocampus plays a pivotal role in supporting memory processes, especially spatial memory (Whitlock et al., 2008). As the initial step in our study, to assess the impact of oA$\beta_{1-42}$ I.C.V. administration on spatial memory, we conducted a Barnes maze task, comprising a specialised evaluation of this form of memory (Gawel et al., 2019). This evaluation was carried out with a first cohort of animals ($n = 11$ per group) (Fig. 3A). The animals underwent training (days 1 and 3) and testing (day 12) in the

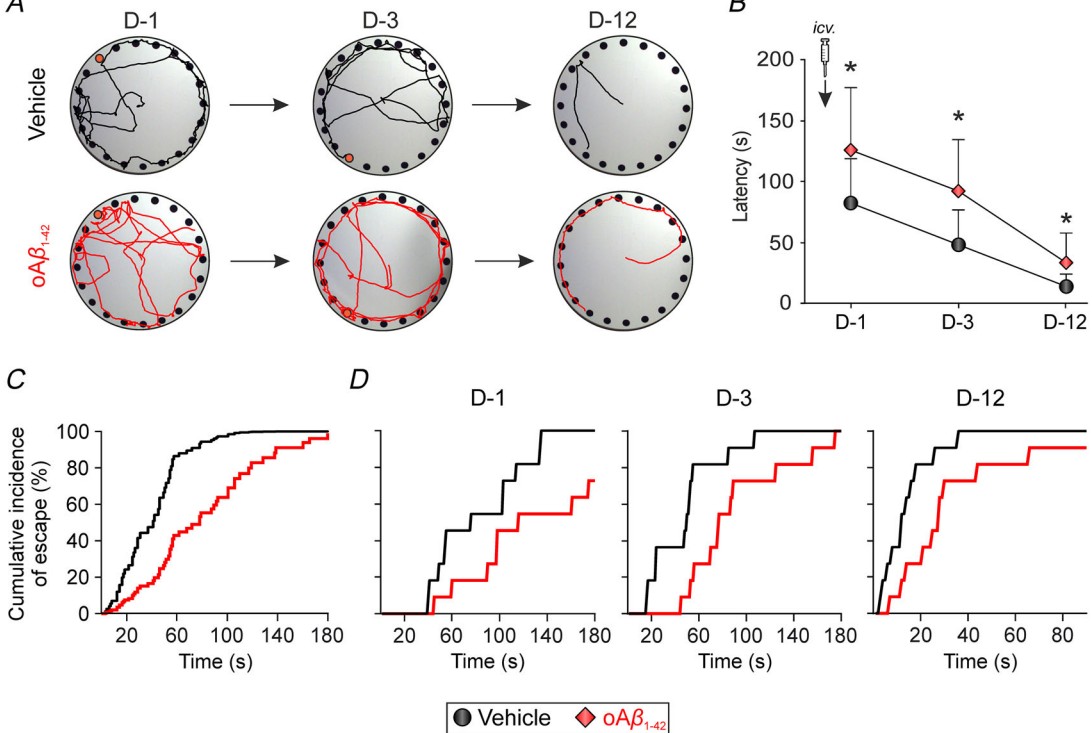

**Figure 3. Alteration of PPC-hippocampal-dependent spatial memory following oA$\beta_{1-42}$ I.C.V. injection**
*A*, representative examples of movement tracking for both experimental groups on each day. *B*, the latency (in seconds) to locate the target hole on different days is provided ($n = 11$ mice per group). Data are expressed as the mean $\pm$ SD of three trials per day. *$P < 0.05$ *vs.* vehicle. *C*, average cumulative incidence plot of finding the escape hole (%) over the duration of the trials. *D*, cumulative incidence plot of finding the escape hole (%) on each training and test day. I.C.V., intracerebroventricular; oA$\beta_{1-42}$, amyloid-$\beta_{1-42}$ oligomers; PPC, posterior parietal cortex.

Barnes maze to evaluate both short-term and long-term spatial memory, as described previously (Jiménez-Herrera et al., 2023). A two-way repeated measures ANOVA unveiled a significant treatment effect ($F_{1,20} = 10.85$, $P = 0.0036$) (Fig. 3*B*) because the oA$\beta_{1-42}$ group displayed an elevated latency to locate the escape hole on all tested days (post-hoc analysis day 1: $P = 0.0145$; day 3: $P = 0.0147$; day 12: $P = 0.0276$). Furthermore, a Cox proportional hazard model was applied to estimate the extent of the effect that the treatment had on the probability of successfully completing the task (i.e. finding the escape hole) (Jahn-Eimermacher et al., 2011). The data showed that both treatment and time had a significant effect on the success rate (Fig. 3*C* and *D*), with the probability of reaching the escape hole being decreased by 72% after oA$\beta_{1-42}$ administration [hazard ratio = 0.28, 95% confidence interval (CI) = 0.158–0.497, $P < 0.001$]. Moreover, across all groups, this rate improved by 26.3% per day (hazard ratio = 1.263, 95% CI = 1.181–1.352, $P < 0.001$). These findings suggest that a single I.C.V. injection of oA$\beta_{1-42}$ deteriorates spatial memory as early as 1 day post-injection and extends this decline, at least, up to 12 days post-injection, affecting both short-term and long-term memory.

Next, to elucidate the relationship and implications of the PPC neuronal activity in spatial memory and to understand the impact of oA$\beta_{1-42}$ I.C.V. administration on this region, we conducted measurements of cortical oscillatory activity (Fig. 4*A* and *B*) using animals from the same cohort across the same time points (pre-treatment, and days 1, 3 and 12 post-treatment). We performed a systematic spectral analysis employing FFT in the frequency domain on LFP recordings ($n = 10$ mice per group, $N_E = 180$ epochs per group, of 10 s each).

In Fig. 4*A* and *B*, the averaged power spectra [$n = 180$ individual spectra per treatment; *i.e.* 10 independent measurements (one for each mouse) with 18 different observations each] of the raw LFP activity recorded during these 4 days is depicted, with the fundamental harmonic centred at 8 Hz in the $\theta$ frequency band (4–12 Hz) for both groups and on all days (analysis intracondition/interbands: $P < 0.01$; Tukey–Kramer test for multiple pairwise comparison of the spectral power between $\theta$ rhythm and the rest of the analysed brain oscillations). Subsequently, a more comprehensive statistical analysis (interconditions/intraband) of the following frequency bands was conducted: $\delta$ (1–4 Hz), $\Theta$ (4–12 Hz), $\beta$ (12–30 Hz), low $\gamma$ (30–48 Hz) and high $\gamma$ (52–150 Hz), represented as a percentage of the pre-treatment session (Fig. 4*D–H*). Normalising to each group's pre-injection baseline allows for a longitudinal assessment of oscillatory activity changes induced by treatment, facilitating a consistent evaluation of within-group dynamics over time.

Graphical representation of the data revealed that in the vehicle group, the spectral powers of both $\delta$ (Fig. 4*D*) and $\theta$ (Fig. 4*E*) rhythms progressively decreased during the learning process (days 1 and 3 post-treatment) and almost returned to baseline (pre-treatment) levels on the test day (day 12 post-treatment). However, the oA$\beta_{1-42}$-treated mice exhibited this decline in spectral power only on the first day of the learning process, after which (days 3 and 12 post-treatment) the spectral powers of both rhythms ($\delta$ and $\theta$) increased. Two-way repeated measures ANOVA revealed a significant treatment effect for both $\delta$ ($F_{1,18} = 13.57$, $P = 0.0017$) and $\theta$ ($F_{(1,18)} = 9.60$, $P = 0.0062$) rhythms.

Regarding the graphical representation of the spectral powers of $\beta$ (Fig. 4*F*) and low (Fig. 4*G*) and high (Fig. 4*H*) $\gamma$ rhythms, these remained stable in the vehicle group throughout the days analysed. Conversely, akin to the observed patterns in slow-wave rhythms ($\delta$ and $\theta$), the oA$\beta_{1-42}$ group showed an increase of these oscillations on days 3 and 12 post-treatment. Two-way repeated measures ANOVA showed a significant treatment effect for $\beta$ ($F_{1,18} = 10.75$, $P = 0.0042$), low $\gamma$ ($F_{1,18} = 7.79$, $P = 0.012$) and high $\gamma$ ($F_{1,18} = 5.19$, $P = 0.035$) spectral powers.

Furthermore, the dynamic evolution of LFP spectral powers was assessed in the vehicle and oA$\beta_{1-42}$-treated groups (Fig. 5) throughout the selected days before (PRE) and after (D-1, D-3 and D-12) the treatment (Fig. 5*B* and *C*). Time-frequency spectrograms of the LFPs were obtained through 18 tapered FTs, with each corresponding to the average of $N_E$ (6 epochs) $\times$ $K$ (3 tapers) in each time step of the spectrogram. These transformations had a frequency width of 1 Hz and a moving time-window of 2 s, shifted in $dt = 500$ ms increments. Matrix of inferences (types +1, 0 and −1) was obtained at each time step ($dt = 500$ ms covering a 300 s epoch) and at each frequency step ($dF = 0.5$ Hz covering the 4–12 Hz band), which was consistent with $2N_E \times K$ degrees of freedom [DOF = 36, $K = 3$ tapers, $N_E = 6$ epochs, one for each animal ($n = 6$ mice per experimental condition (days))]. However, treating each time step in a spectrogram as an independent measure is not appropriate from a comprehensive statistical perspective as successive 500 ms increments may exhibit autocorrelation (i.e. the value at a given time depends on the values at previous time steps), implying that linear independence between successive time windows cannot be guaranteed. Although the mTFT approach was suitable for generating descriptive estimates of the power spectrograms, it is not valid for inferential comparisons between groups because the number of animals should remain as the true independent measurement ($n = 6$ mice for mTFT analysis). According to the above, the results of Fig. 5 are presented through a descriptive statistical analysis based on pie chart

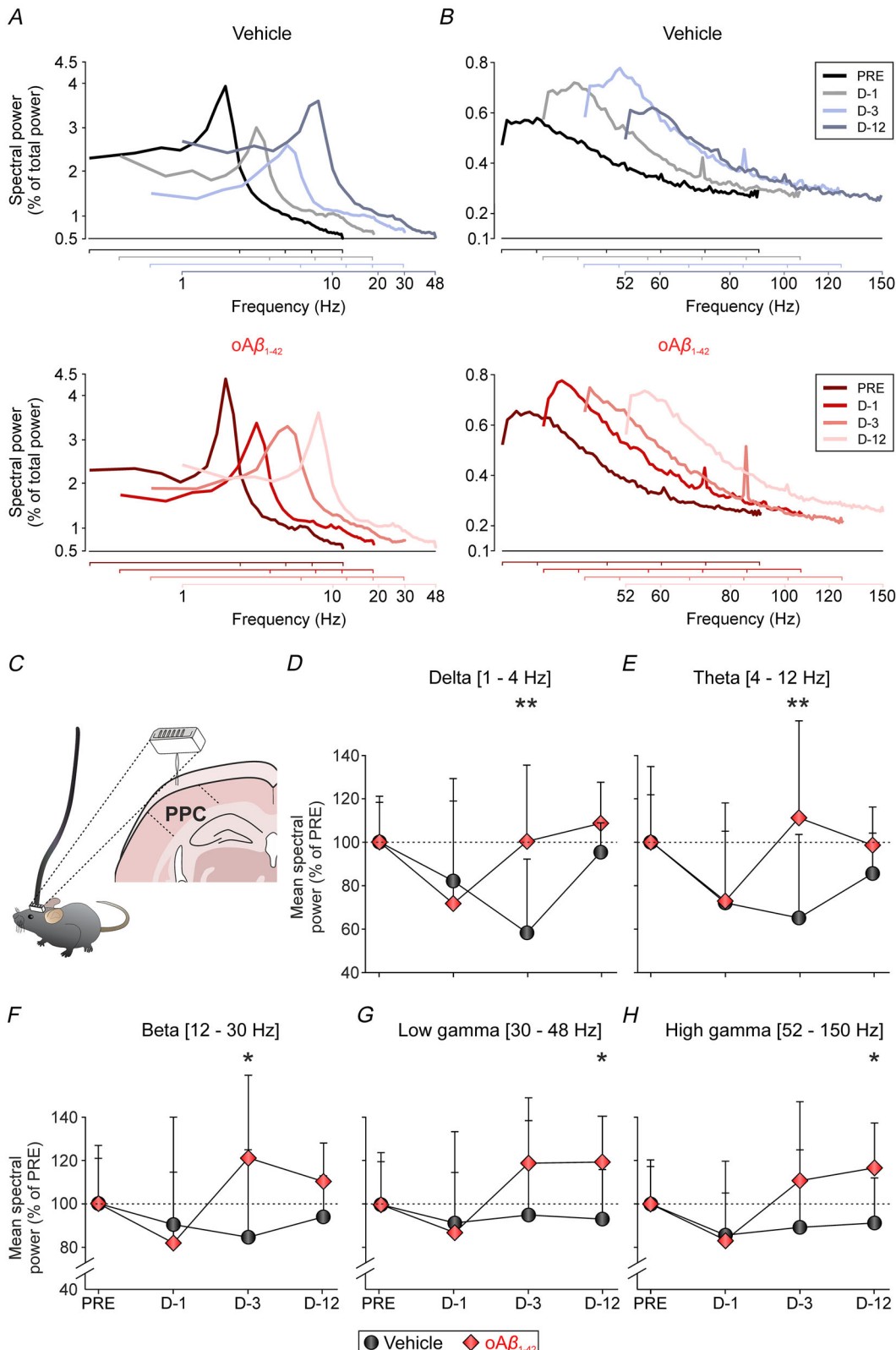

**Figure 4. Alteration of PPC oscillatory activity following oAβ_{1-42} I.C.V. injection**
*A* and *B*, averaged power spectra from 10 s epochs covering 3 min of recordings (18 epochs) acquired from
*n* = 10 mice per group (*N*_E = 180 epochs) divided into (*A*) low- and (*B*) high-frequency bands for both vehicle (up,
grey scale) or oAβ1-42 (down, red scale). Recordings were collected on day 0 previous to I.C.V. administration (PRE)
and on days 1, 3 and 12 post-I.C.V. injections (D-1, 3 and 12). Data are expressed as a percentage (%) of total

spectral power for visual representation purposes. *C*, graphical representation of the electrode placement in the PPC. *D–H*, mean ± SD spectral power for different frequency bands analysed is represented. *D*, delta ($\delta$, 1–4 Hz). *E*, theta ($\theta$, 4–12 Hz). *F*, beta ($\beta$, 12–30 Hz). *G*, low gamma ($\gamma$, 30–48 Hz). *H*, high $\gamma$ (52–150 Hz). Data are represented as a percentage of the pre-injection day (100%). Significance level comparing $oA\beta_{1-42}$ *vs.* vehicle is reported as: *$P < 0.05$ (Tuker–Kramer test for the mean powers from both D-3 and D-12). I.C.V., intracerebroventricular; $oA\beta_{1-42}$, amyloid-$\beta_{1-42}$ oligomers; PPC, posterior parietal cortex.

representations for the probability density of inferences [types ±1 ($P < 0.05$) and type 0 ($P > 0.05$)].

The results collected allowed to confirm that the maximum spectral power consistently appeared in the $\theta$ band (4–12 Hz, centred at 8 Hz) across all cases ($\theta$ *vs.* all other frequency bands at all time points in both experimental groups, $P < 0.01$). However, there were varying degrees of contribution to the $\theta$ fundamental rhythm. Figure 5*A* and *D* depicts pie charts of the probability density resulting from multiple comparisons between pairs of spectrograms. Black portions [indicating type +1 inference; *E*2nd (estimation of power in second) ≫ *E*1st (estimation of power in first)] and grey portions (indicating type −1 inference; *E*2nd ≪ *E*1st) denote significant statistical differences ($P < 0.05$; Jackknifed estimates of the variance) in mean spectral powers. White portions of the pie chart (indicating type 0 inference; *E*2nd ≈ *E*1st) signify no significant differences ($P > 0.05$). In descriptive terms, pie charts in Fig. 5*A* and *D* (left) reveal that spectral powers of LFPs recorded in PPC during the first day after the injection were lower than those obtained before the injection [vehicle: D-1 *vs.* PRE: types +1 (47.4%) and 0 (45.5%) inferences were predominant); $oA\beta_{1-42}$: D-1 *vs.* PRE: types +1 (54.2%) and 0 (42.4%) inferences were predominant)] for the $\theta$ frequency band. The differences in the probability density were even more noticeable when comparing spectral powers between the three selected days following injection, in both the $oA\beta_{1-42}$ [D-1 *vs.* D-3: types +1 (72.5%) and 0 (25.9%) inferences were predominant; D-1 *vs.* D-12: types +1 (52.2%) and 0 (44.6%) inferences were predominant] and the vehicle groups [D-1 *vs.* D-3: type 0 (53.3%) inferences were predominant; D-1 *vs.* D-12: type 0 (57.8%) inferences were predominant].

To summarise, for both experimental groups, the spectral powers corresponding to $\theta$ rhythm significantly decreased on the first day after injection (D-1 compared to the PRE day). However, in $oA\beta_{1-42}$-treated mice, there was a noticeable increase in $\theta$-band spectral powers on the third day (D-3) after injection, in contrast to that observed for LFPs from the PPC of vehicle mice whose spectral powers decreased. Finally, by the time day D-12 (post-injection) was reached, the power values were quite close to those on the PRE-injection day.

Additionally, a cross-frequency comodulation analysis has been performed, as described elsewhere (Masimore et al., 2004). For both experimental groups, $\theta–\gamma$ comodulation was not observed in the LFP recordings across different days (PRE, D-1, D-3 and D-12) (Fig. 6). This was the case despite a noticeable increase in LFP spectral powers of both $\theta$ and $\gamma$ in $oA\beta_{1-42}$-treated mice during the days D-3 and D-12 after injection [relative to pre-injection day (PRE)] (Fig. 4*A*, *B* and *D–H*). The parallel increase in $\theta$ and $\gamma$ spectral powers did not lead to significant $\theta–\gamma$ coupling in PPC LFP recordings. This suggests that the observed spectral pattern contributes more to the modulation of power within each frequency band rather than to cross-frequency comodulation between slow and fast oscillations. However, our findings did identify moderate-to-weak comodulations (from 0.5 to 0.2, see colour bar) between low-frequency rhythms; specifically, $\theta–\beta$ interactions (on day PRE and on day D-12 post-injection) and $\delta–\beta$ interactions (on day D-3 post-injection).

Surprisingly, spatial memory deficits were detectable from the first day post-treatment (D-1), even though the oscillatory activity of the PPC was not affected until the third day post-injection (D-3). These data suggest a potential impairment in hippocampal functionality induced by early amyloidosis, as previously demonstrated by our group and others (Jiménez-Herrera et al., 2023; Kim et al., 2016), at an earlier stage than the alteration of the PPC's network activity.

### Alterations in hippocampal memory, oscillations and synaptic plasticity induced by $oA\beta_{1-42}$ I.C.V. injection within 24 h

To further investigate the early impact of amyloidosis on hippocampal functionality, we conducted experiments with a second cohort of animals, aiming to assess the effects of $oA\beta_{1-42}$ I.C.V. administration on hippocampal-dependent memory, hippocampal oscillatory activity and long-term synaptic plasticity (Fig. 7).

In this instance, we chose to evaluate habituation memory to a novel environment, a type of memory closely related to hippocampal function (Leussis & Bolivar, 2006). To achieve this, we utilised an OF habituation test, which also enables the assessment of shorter time periods, facilitating the exploration of hippocampal functionality in the early stages of amyloidosis-induced alterations. One hour after treatment, animals ($n = 11$ mice per group) were placed in an OF (OF1, training session) for 15 min. During this session, one-way repeated measures

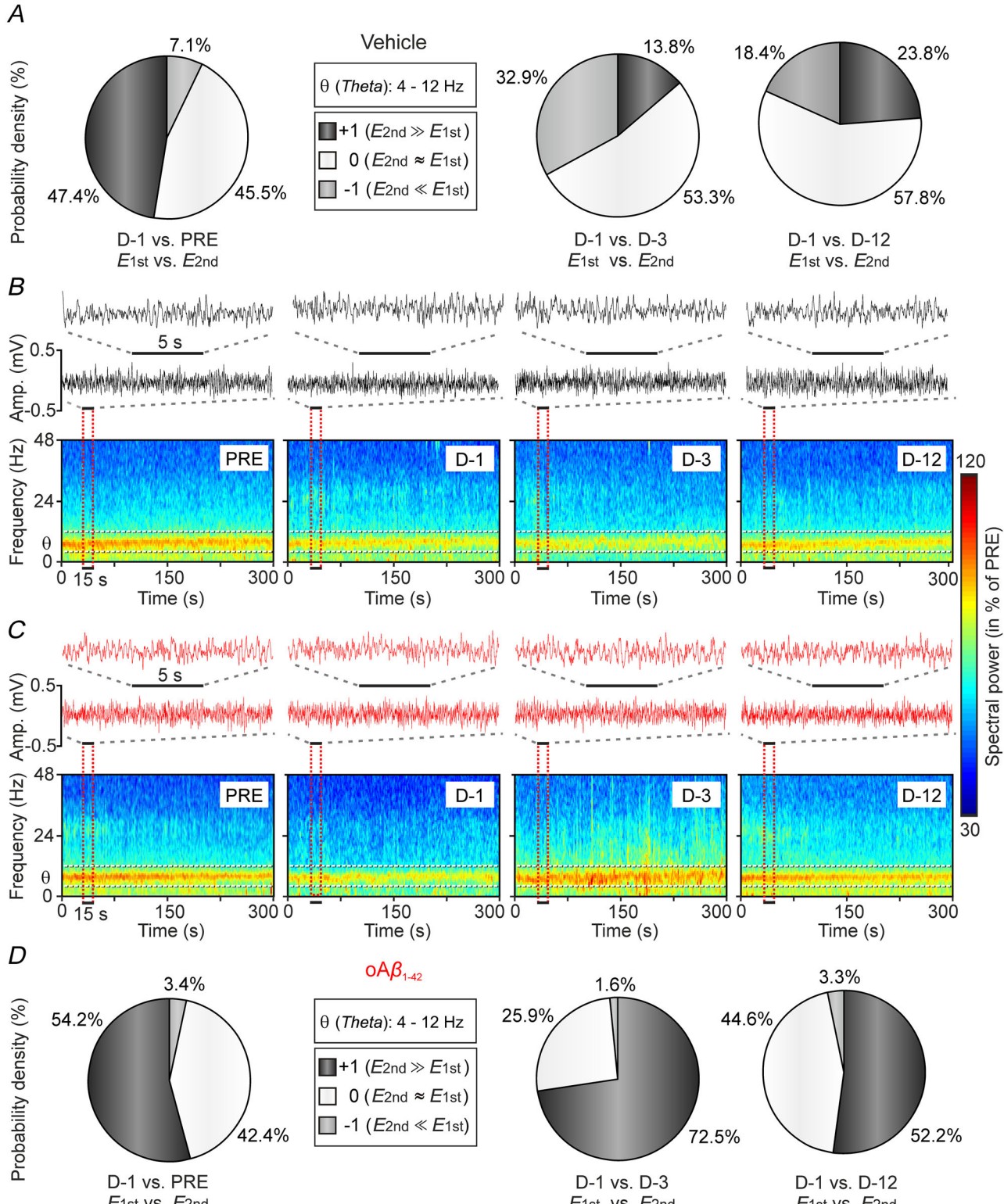

**Figure 5. PPC theta (θ) rhythm spectral dynamics based on multitapered Fourier transform (mTFT)**
*A*, pie charts representing the mean probability densities for the group of vehicle mice (*n* = 6). In the D-1 *vs*. PRE comparison, both type +1 (black bar) and type 0 (white bar) inferences were predominant. However, in the D-1 *vs*. D-3 and D-1 *vs*. D-12 comparisons, only a predominance of type 0 inferences is observed (white bars; *E*2nd ≈ *E*1st; ****P* < 0.001) *B*, time–frequency representations (spectrograms; $N_E \times K$ = 18 tapered Fourier transforms for each time step) corresponding to LFPs recorded from vehicle mice (*n* = 6) across different days (PRE, D-1, D-3 and D-12).

Horizontal dashed lines indicate the $\theta$ (4–12 Hz) band. *C*, spectrograms corresponding to LFPs recorded from oA$\beta_{1-42}$ mice (as presented in *B*), which also exhibit a pronounced drop in spectral power on day D-1 compared to day PRE. These spectrograms ($K = 3$ tapers, $N_E = 6$ epochs per group of 300 s of duration each) allow visualisation of the time–frequency evolution of spectral powers and confirm the descriptive information represented in (*A*) and (*D*). *D*, pie charts depicting the mean probability densities for the group of oA$\beta_{1-42}$ mice ($n = 6$). The three comparisons (D-1 *vs.* PRE; D-1 *vs.* D-3; D-1 *vs.* D-12) reveal a predominance of type +1 inferences (black bars; $E$2nd $\gg E$1st; ***$P < 0.001$). This indicates a significant decrease in spectral power in the $\theta$ band on the first day after the injection compared to the previous day (PRE) and days D-3 and D-12 post-injection. LFP, local field potential; oA$\beta_{1-42}$, amyloid-$\beta_{1-42}$ oligomers; PPC, posterior parietal cortex.

ANOVA showed that the exploratory activity of both groups progressively reduced (vehicle: $F_{14\,140} = 7.89$, $P < 0.001$; oA$\beta_{1-42}$: $F_{14\,140} = 19.48$, $P < 0.001$) (Fig. 7*A*), indicating proper encoding of habituation memory. No significant treatment effect was observed at this stage ($F_{1,20} = 0.18$, $P = 0.679$). Twenty-four hours after i.c.v. treatment, animals were reintroduced into the OF arena (OF2, recall session). Here, a significant decrease in exploratory activity was observed in the control group ($t_{10} = 4.36$, $P = 0.0015$) (Fig. 7*B*), indicating their ability to remember the arena. By contrast, this habituation effect was not observed in oA$\beta_{1-42}$-treated mice ($t_{10} = 0.30$, $P = 0.774$) (Fig. 7*B*), suggesting a deterioration in memory recall for this group. Furthermore, during this recall session (OF2), two-way repeated measures ANOVA revealed a significant treatment, with oA$\beta_{1-42}$-treated mice covering a longer distance than vehicle mice ($F_{1,20} = 6.51$, $P = 0.019$) (Fig. 7*B*). As all animals,

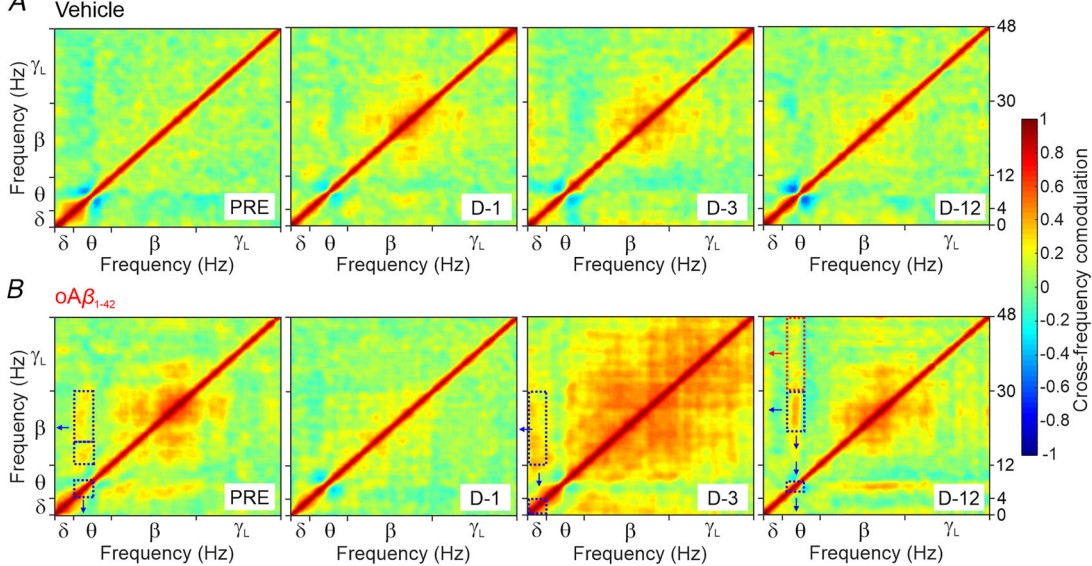

**Figure 6. Cross-frequency comodulation analysis for LFPs from the PPC region**
These frequency–frequency representations show the interaction strength (in a statistical sense, see colour bar) and spectral pattern between two oscillations derived from LFP recordings. *A*, comodulograms for LFPs (300 s epochs) recorded from vehicle mice ($n = 6$) reveal no significant interactions between pairs of low-frequency oscillations [delta ($\delta$), $\theta$ and beta ($\beta$) bands]. *B*, comodulograms for LFPs from oA$\beta_{1-42}$ mice ($n = 6$) (as presented in *A*) across different days (PRE, D-1, D-3 and D-12) exhibit a weak comodulation between low $\theta$ (4–8 Hz, 0.444 ± 0.039, 95% CI = 0.365–0.522) and $\beta$ (low, 12–18 Hz, 0.200 ± 0.007, 95% CI = 0.187–0.213; high 18–30 Hz, 0.195 ± 0.005, 95% CI = 0.185–0.204) bands on day PRE (see dashed rectangle and blue arrows) compared to the first day (D-1) post-injection, where cross-frequency coupling was not significant (much smaller than 0.2) outside the diagonal. Moderate comodulation between $\delta$ (0–4 Hz, 0.611 ± 0.028, 95% CI = 0.555–0.666) and $\beta$ (12–30 Hz, 0.265 ± 0.004, 95% CI = 0.258–0.271) oscillations was observed on the day D-3, contrasting with moderate comodulation observed between low $\theta$ (range 6–8 Hz, 0.791 ± 0.063, 95% CI = 0.646–0.936) and high $\beta$ (18–30 Hz, 0.322 ± 0.007, 95% CI = 0.307–0.337) rhythms on day D-12 post-injection. The $\theta$–$\gamma$ couplings were not detected in both animal groups with this method. For example, the comodulation between low $\theta$ (6–8 Hz, 0.791 ± 0.063, 95% CI = 0.646–0.936) and low $\gamma$ (30–48 Hz, 0.187 ± 0.007, 95% CI = 0.174–0.201) rhythms on day D-12 post-injection (see red dashed rectangle) was not statistically significant (rejection of the null hypothesis that mean values between the groups are equal: $H = 24.7$, $P < 0.001$). Ranges: 0–0.2, weak comodulation; 0.2–0.5, moderate comodulation; 0.5–1.0 high comodulation. CI, confidence interval; LFP, local field potential; oA$\beta_{1-42}$, amyloid-$\beta_{1-42}$ oligomers; PPC, posterior parietal cortex.

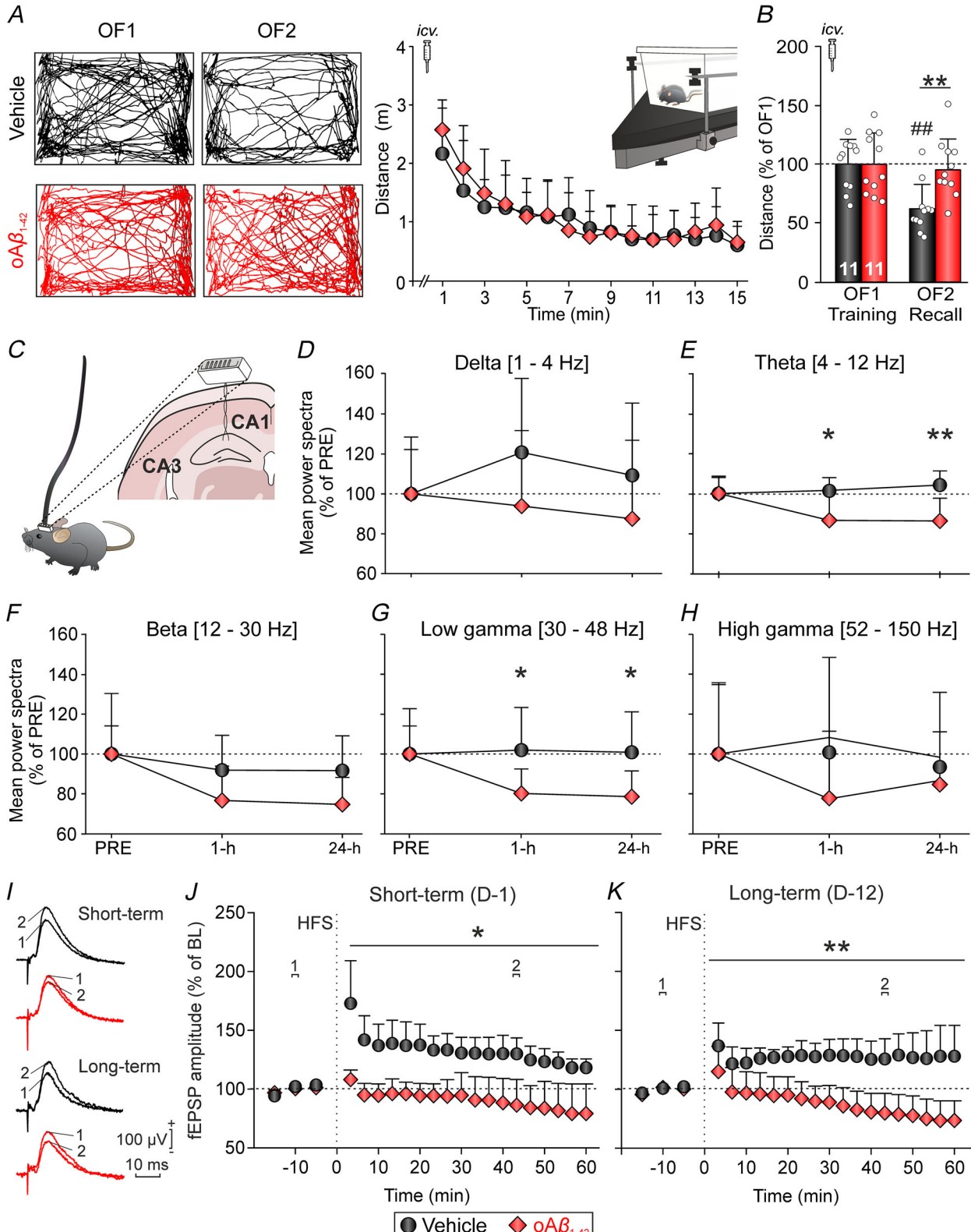

**Figure 7. Disruption of hippocampal-dependent memory, oscillation activity and synaptic plasticity following I.C.V. administration of oA$\beta_{1-42}$**

*A*, memory acquisition was assessed by calculating the distance travelled by the animals during each min of the OF1 session (right). Representative examples of movement tracking for both experimental groups are illustrated on the left. Data are expressed as the mean ± SD. *B*, total distance travelled by each experimental group (*n* = 11 mice

per group) during both training and recall sessions is expressed as a percentage of the distance travelled during OF1 (set as 100%). \*\*$P < 0.01$ *vs.* vehicle; ##$P < 0.01$ *vs.* OF1. *C*, graphical representation of the electrode placement in the CA1 area. *D–H*, spectral power (mean ± SD) of different frequency bands recorded from the hippocampus ($n = 8$ mice per group) 1 h and 1 day post-treatment. *D*, delta ($\delta$, 1–4 Hz). *E*, theta ($\theta$, 4–12 Hz). *F*, beta ($\beta$, 12–30 Hz). *G*, low gamma ($\gamma$, 30–48 Hz). *H*, high $\gamma$ (52–150 Hz). Data are represented as a percentage of the pre-injection day (100%). \*$P < 0.05$ *vs.* vehicle. *I*, representative averaged traces ($n = 40$) of fEPSPs recorded in the CA1 area, collected during the baseline (1) and ≈45 min post-HFS (2) in hippocampal slices from different experimental groups. *J* and *K*, time course of LTP evoked in the CA1 area after an HFS session in hippocampal slices after (*J*) 1 day (short term; $n = 6$ slices, 3 mice vehicle, 7 slices, 3 mice oA$\beta_{1-42}$) and (*K*) 12 days (long term; $n = 8$ slices, 4 mice vehicle, 8 slices, 3 mice oA$\beta_{1-42}$) post-vehicle or oA$\beta_{1-42}$ injections. \*$P < 0.05$, \*\*$P < 0.01$ *vs.* vehicle. HFS, high frequency stimulation; I.C.V., intracerebroventricular; LTP, long-term potentiation; oA$\beta_{1-42}$, amyloid-$\beta_{1-42}$ oligomers; OF, Open field.

regardless of treatment, decreased exploration along the training session (Fig. 7*B*, left), but only oA$\beta_{1-42}$-treated mice showed impaired retrieval during recall testing (Fig. 7*B*, right), it could be hypothesised that, in addition to retrieval, storage and or consolidation processes, more probable than encoding, could also be impaired by oA$\beta_{1-42}$. In summary, these data suggest that early amyloidosis clearly impairs exploratory habituation memory retrieval, and these alterations occur within the first 24 h post-injection.

Having established the impact of amyloidosis on hippocampal-dependent memory, our next objective was to investigate the role of hippocampal neuronal activity in these alterations. Thus, we analysed the oscillatory activity of the CA1 hippocampal region at different time points during the OF habituation task (pre-I.C.V. injections and 1 and 24 h post-treatment). To achieve this, recording electrodes were implanted in the CA1 area of the mice's brains ($n = 8$ mice per group) (Fig. 7*C*) and the same frequency bands as previously analysed in the PPC were examined, represented as a percentage of the pre-treatment session. The results revealed that in mice treated with oA$\beta_{1-42}$, a decrease in the spectral power was observed in all analysed rhythms starting from 1 h post-injection and persisting, at least, up to 24 h post-injection, compared to the control animals (Fig. 7*D–H*). Two-way repeated measures ANOVA showed that this decrease as a result of treatment was statistically significant in both $\theta$ ($F_{1,14} = 11.65$, $P = 0.0042$) (Fig. 7*E*) and low $\gamma$ ($F_{1,14} = 10.56$, $P = 0.0058$) (Fig. 7*G*) rhythms. This suggests that I.C.V. injection of oA$\beta_{1-42}$ can lead to alterations in hippocampal oscillatory activity as early as 1 h post-injection, well before the observed effects in the PPC.

Finally, because hippocampal-dependent memory relies not only on oscillatory activity, but also on synaptic plasticity (Bliss et al., 2018), a subset of mice from the second cohort of animals (cohort 2B) was used to evaluate hippocampal LTP *ex vivo*. To match the potential effects with those observed in the *in vivo* recordings and behavioural tasks, *ex vivo* electrophysiology was carried out 1 (short-term; $n = 6$ slices vehicle, 7 slices oA$\beta_{1-42}$) (Fig. 7*I* and *J*) and 12 days (long-term; $n = 8$ slices vehicle,

8 slices oA$\beta_{1-42}$) (Fig. 7*I* and *K*) post-I.C.V. injection. In both time points, although, in slices from vehicle-treated mice, the HFS protocol resulted in ~130% potentiation of the response evoked in CA1, in slices from mice treated with oA$\beta_{1-42}$, the induction of LTP was hindered, leading to a depression of the fEPSP (~80% of the baseline) in response to the same HFS protocol (two-way repeated measures ANOVA treatment effect: short-term: $F_{1,11} = 28.96$, $P = 0.0002$; long-term: $F_{1,14} = 45.56$, $P < 0.0001$).

These results indicate the hippocampus's special vulnerability to oA$\beta_{1-42}$ I.C.V. administration because physiological hippocampal deficits manifest as early as 1 h post-treatment, preceding the observed effects in the PPC. The subsequent reflection of alterations in the hippocampal synaptic network in the PPC hints at the potential involvement of both regions in spatial memory processing, underscoring the importance of their proper functioning for optimal cognitive performance. Altogether, these findings enhance our understanding of the temporal progression of A$\beta$ peptide's deleterious effects in the early stages of amyloidosis.

### Lack of effects of oA$\beta_{1-42}$ I.C.V. injection on locomotion, energetic state and emotional well-being

To ensure that the observed alterations following oA$\beta_{1-42}$ injection were indeed related to specific impairments in the PPC and the hippocampus rather than general health issues, a battery of behavioural tests was conducted to evaluate various behavioural aspects.

Stereotyped behaviours were assessed and LABORAS data revealed that all experimental groups spent equivalent amounts of time engaged in the various behaviours analysed ($n = 10$ mice per group) (Fig. 8*A*): locomotion ($U = 47$, $P = 0.821$), climbing ($U = 46$, $P = 0.691$), rearing ($U = 39$, $P = 0.374$) and grooming ($U = 40.5$, $P = 0.471$). Using an OF, no disparities were observed in the distance travelled ($t_{18} = 0.28$, $P = 0.785$) (Fig. 8*B*), a measure of motor activity, nor in the time spent in the periphery ($t_{18} = 0.266$, $P = 0.793$, $n = 10$ mice per group) (Fig. 6*B*), indicating that the animals did not exhibit anxiety-like behaviours. Locomotion was

further assessed through the number of entries into closed ($t_{17} = 0.14$, $P = 0.887$) and total arms ($t_{17} = 0.21$, $P = 0.834$) in the elevated plus maze and, once again, with no differences found between groups ($n = 10$ mice vehicle, 9 mice $oA\beta_{1-42}$) (Fig. 8C). Moreover, all animals displayed similar numbers of entries ($U = 37$, $P = 0.49$) and time spent in open arms ($U = 40$, $P = 0.671$) in the elevated plus maze (Fig. 6C), reinforcing that induced amyloidosis did not elevate anxiety-like behaviours. One-way repeated measures ANOVA revealed that all groups equally improved their performance in the rotarod across trials (vehicle: $F_{5,45} = 3.388$, $P = 0.011$; $oA\beta_{1-42}$: $F_{5,35} = 3.926$, $P = 0.006$, $n = 9$ mice vehicle, 8 mice $oA\beta_{1-42}$) (Fig. 8D) and there were no differences in the latency to fall off the rod during the entire session (two-way repeated measures ANOVA: $F_{1,16} = 0.005$, $P = 0.943$) (Fig. 8D), indicating no disparities in motor co-ordination as a result of treatment. Finally, all animals exhibited the same levels of energy ($t_{18} = 0.829$, $P = 0.418$) and power of movement ($t_{18} = 0.216$, $P = 0.831$), along with identical immobility times ($t_{18} = 0.954$, $P = 0.353$) in the tail suspension test ($n = 10$ mice per group) (Fig. 8E). This suggests that $oA\beta_{1-42}$ did not induce muscle weakness or increase depression-like behaviours.

In summary, these results confirm that the overall health state, locomotor function and emotional state remained consistent across both treated groups. Therefore, the impairments observed can be attributed to a specific disruption in the PPC and/or hippocampal circuitry caused by $oA\beta_{1-42}$.

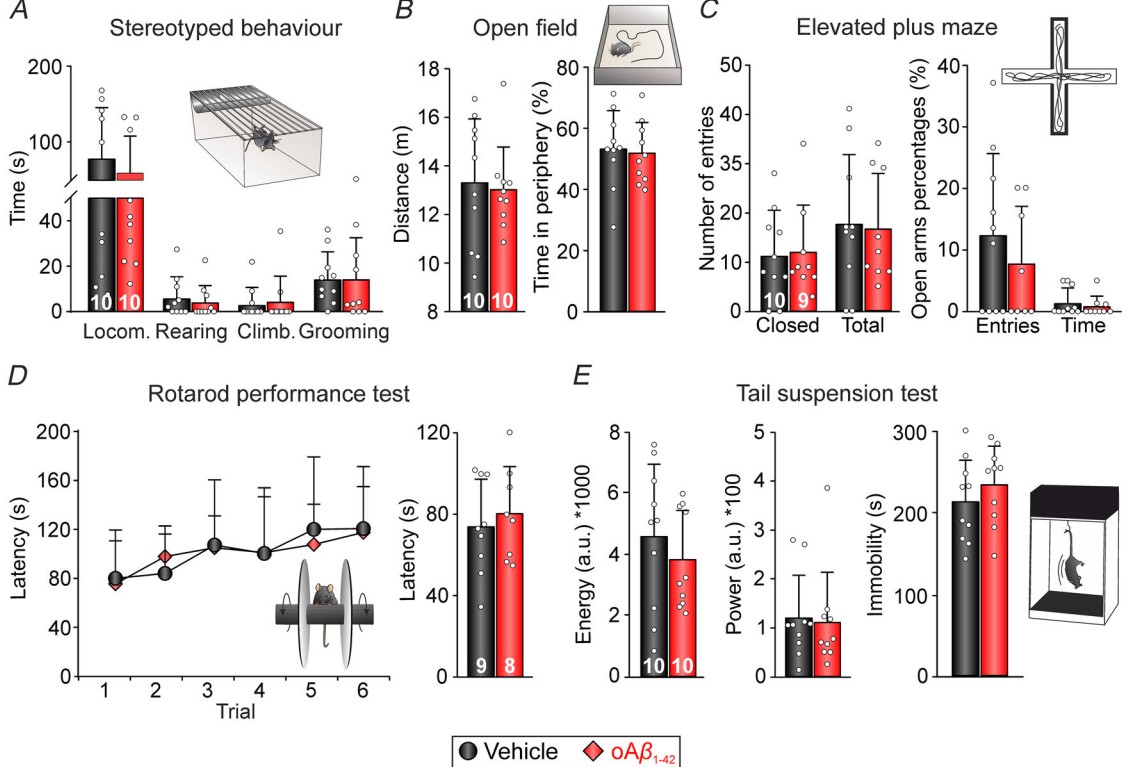

**Figure 8. Effects of I.C.V. injection of $oA\beta_{1-42}$ on motor function and anxiety-like behaviours**
*A*, various spontaneous behaviours observed during a single 15 min LABORAS session were evaluated based on the vibration patterns generated by the animals ($n = 10$ mice per group). The bar plots illustrate the time spent by the animals engaging in locomotion (locom.), rearing, climbing (climb.) or grooming. *B*, locomotion was quantified by measuring the distance travelled in an open field arena during a single 15 min LABORAS session (left), whereas anxiety levels were assessed by determining the percentage of time the animals spent on the periphery of the arena (right; $n = 11$ mice per group). *C*, locomotor activity was further assessed using an elevated plus maze by measuring the number of entries into closed arms or total arms during a single 5 min session (left). This test also served to evaluate anxiety, as indicated by the percentage of open arm entries and the percentage of time spent in open arms (right; $n = 10$ mice vehicle, 9 mice $oA\beta_{1-42}$). *D*, motor function and co-ordination were evaluated using a rotarod, where the latency to fall off the rod was measured for each trial (left) and for the entire session (right; $n = 10$ mice vehicle, 9 mice $oA\beta_{1-42}$). *E*, energy levels (left) and the power of movements (middle) were automatically assessed during a single 6 min session in the tail suspension test. Immobility time was quantified as a measure of depression-like behaviour (right; $n = 11$ mice per group). The number of animals of each group is indicated on the corresponding bar. LABORAS, Laboratory Animal Behavior Observation Registration and Analysis System; I.C.V., intracerebroventricular; $oA\beta_{1-42}$, amyloid-$\beta_{1-42}$ oligomers.

## Discussion

In the early stages of AD, imbalances in neuronal activity within various brain circuits, including the cortex and hippocampus, have been consistently observed in both patients and murine models (Busche & Konnerth, 2015; Vossel et al., 2013). Evidence supports the role of soluble $oA\beta$ in contributing to the cognitive deficits commonly associated with these altered neuronal states seen during the preclinical stages of the disease (Palop & Mucke, 2010). Previous research from our group has delved into the impact of hippocampal amyloidosis induced by oligomeric forms of $A\beta_{1-42}$, revealing increased excitability that coincides with disrupted LTP, irregular patterns in hippocampal oscillatory activity and deficits in hippocampal-dependent memory that persists over time (Djebari et al., 2025; Jiménez-Herrera et al., 2023; Mulero-Franco et al., 2025; Sánchez-Rodríguez et al., 2017, 2019, 2020).

To gain a deeper understanding of how $A\beta$ influences cognitive processes, particularly hippocampal-dependent memory, it becomes crucial to consider other regions with which the hippocampus establishes connections and understand how $A\beta$ deleterious action progresses. Thus, the present study has placed particular emphasis on investigating the time sequence between the hippocampus and the PPC impairments because these regions are known to synchronise their neuronal activity to support functions such as spatial memory formation or episodic memory retrieval (Sestieri et al., 2017; Whitlock et al., 2008). To initiate our exploration, we conducted a Barnes maze test to assess spatial memory. Our findings revealed a significant increase in the latency to find the target hole, as well as a decrease in the probability of reaching this hole, 24 h after ɪ.ᴄ.ᴠ. administration of $oA\beta_{1-42}$ peptide. This decline in spatial navigation memory persisted for at least 12 days after amyloidosis induction. These results have dual implications. First, they suggest that oligomeric forms of $A\beta$ could be affecting the functionality of the PPC and the hippocampus, both crucial for spatial memory (Burke et al., 2005; Whitlock et al., 2008). This behavioural impairment is consistent with our previous research, which demonstrated that acute amyloidosis induced by a single injection of $oA\beta_{1-42}$ impairs long-term spatial memory (Jiménez-Herrera et al., 2023). Furthermore, it aligns with reports indicating a compromised spatial navigation in early phases of the disease (Coughlan et al., 2018; Lithfous et al., 2013). Second, these findings confirm the suitability of our amyloidosis model to mimic the early stages of AD, previously validated in both male and female mice without finding differences between the two (Jiménez-Herrera et al., 2023). This previous validation underscores the model's reliability and sex independent applicability, enabling us to use animals of either sex in subsequent research confidently. Such a strategy not only

ensures greater sample homogeneity, but also aligns with the 3Rs principle by minimising the number of animals required. However, it remains essential to underline the significance of incorporating animals of both sexes in future studies, particularly those testing new drugs, given the potential for sex-specific effects that have yet to be examined.

Subsequently, to explore the role of PPC functionality in this process, we surgically implanted subdural electrodes in the PPC to record LFPs in alert mice. Oscillatory activity of neural networks arises from the activity of different neuronal populations, and their synchronisation defines the amplitude (or spectral power) of oscillations at different frequency bands. In this context, GABAergic inhibitory interneurons play a pivotal role in co-ordinating the activity and synchronisation of excitatory neurons and other interneurons, thereby generating oscillations across different frequency bands (Palop & Mucke, 2016). Such oscillatory activity is ultimately linked to spatial memory, changing during both navigation (Chrastil et al., 2022) and retrieval of spatial memories (Vivekananda et al., 2021). To avoid disrupting the integrity of behavioural test results, LFP recordings in this study were performed following the behavioural tests, leaving the influence of test conduct on oscillatory activity outside the scope of our examination. However, given that dysfunction within neuronal networks manifests as alterations of oscillatory activity across different frequency intervals associated with cognitive processes (Palop & Mucke, 2016; Schnitzler & Gross, 2005), we aimed to assess the impact of $oA\beta_{1-42}$ ɪ.ᴄ.ᴠ. administration on PPC's oscillatory activity. Our LFP recordings revealed that, whereas oscillatory activity in the PPC remained unaltered 24 h after $oA\beta_{1-42}$ ɪ.ᴄ.ᴠ. injection, a significative increase in spectral power across all analysed rhythms emerged at later time points. This effect became evident at 3 days post-ɪ.ᴄ.ᴠ. injection in lower-frequency rhythms ($\delta$, $\theta$ and $\beta$) and was subsequently extended to higher-frequency bands (low and high $\gamma$) by 12 days post-ɪ.ᴄ.ᴠ. injection. This change was particularly pronounced within the $\theta$ rhythm, where the heightened spectral power enabled us to conduct a more comprehensive temporal analysis. This analysis unveiled an aberrant increase in spectral power within the $\theta$ band, starting from the third day post-injection in amyloidosis-affected animals. Consistent with our observations, alterations in $\theta$ and $\gamma$ rhythms have been documented in electroencephalogram (EEG) recordings of AD patients (Adler et al., 2003; Herrmann & Demiralp, 2005) and in LFP recordings of murine models of the disease (Stoiljkovic et al., 2019; Wang et al., 2020). This supports the notion that accurate memory encoding relies on these specific rhythms within task- relevant neural networks (Palop & Mucke, 2016). Notably, $\theta$ and $\gamma$ oscillations interact to create the $\theta$–$\gamma$ neural code,

involved in sensory processing, memory and learning (Lisman & Jensen, 2013), a coupling potentially disrupted in the early stages of AD (Zhang et al., 2016). Although this study did not directly address $\theta$–$\gamma$ coupling, preliminary findings based on cross-frequency comodulation suggest no significant couplings between slow and fast oscillations in our model, with somewhat stronger comodulations noted among low-frequency rhythms instead. Nevertheless, more in-depth analysis in this direction presents a compelling pathway for future research. Additionally, EEG recordings from patients in the preclinical stages of AD have also revealed alterations in other cortical rhythms, including $\delta$ and $\beta$ rhythms, which may contribute to an increased risk of future cognitive decline (Gaubert et al., 2019). In addition, these findings align with other studies that describe how neuronal hyperactivity induced by A$\beta$ in the early stages of AD and in murine models of amyloidosis can contribute to the dysfunction of neural networks and trigger epileptiform activity (Jeremic et al., 2021; Zott et al., 2018).

Furthermore, a noteworthy observation that merits further explanation is the temporal dissociation between the emergence of spatial memory deficits (which were evident from the same day post-treatment) and the absence of alterations in PPC oscillatory activity until 3 days post-injection. Neuronal activity within a specific region is significantly influenced by afferent signals it receives from other interconnected brain regions (Herreras, 2016). Thus, the LFPs obtained from the surface of the PPC could reflect changes that have previously occurred in other structurally and functionally interconnected areas, including the hippocampus. In this scenario, the i.c.v. administration of the oA$\beta_{1-42}$ might trigger acute amyloidosis primarily at the hippocampal level, which would be affected initially. As a result, oA$\beta$-induced alterations in hippocampal neuronal activity would interfere early with hippocampal-dependent memory, yet the effects might not manifest in cortical oscillatory activity until several days later, corresponding to the long-term consolidation of this memory type. To examine this hypothesis, we assessed a second cohort of animals with electrodes implanted in dorsal CA1 to analyse short-term changes in hippocampal-dependent memory and hippocampal oscillatory activity.

At the behavioural level, we examined how oA$\beta$-induced hippocampal amyloidosis affects different aspects of hippocampal-dependent memory processing. To do so, we conducted an OF habituation test, a tool that allows us to assess two specific types of habituation processes: intrasession, which evaluates memory formation within the same session, and intersession, which assesses memory recall, across different sessions, both of which dependent largely on the hippocampus (Daenen et al., 2001; Wright et al., 2004). In this

test, both the control group and the mice receiving oA$\beta_{1-42}$ displayed a gradual memory formation curve during the initial session, very probably indicating no impaired encoding of memory. However, they exhibited deficits in habituation memory retrieval 24 h after the injections. These results are consistent with previous studies describing impairments in habituation memory to a new context and object recognition memory in models of early amyloidosis (Jiménez-Herrera et al., 2023; Mayordomo-Cava et al., 2020; Sánchez-Rodríguez et al., 2017, 2020) and substantiate the results observed in the Barnes maze test, which also pointed to memory impairments. It is worth noting that the PPC, renowned for its role in memory retrieval (Sestieri et al., 2017), may be implicated in these observed memory deficits, although, in the present study, we found no evidence of aberrant oscillatory activity in this region until 3 days after induction of amyloidosis. Subsequently, to gain insights into hippocampal functionality and elucidate the underlying factors contributing to the observed hippocampal-dependent memory impairments, we conducted recordings of the oscillatory activity in the CA1 area of the hippocampus. LFPs unveiled a significant reduction in $\theta$ and low $\gamma$ rhythms as early as 1 h following the administration of oA$\beta_{1-42}$ suggesting that alterations in oscillatory activity occur in the hippocampus before the PPC, although recordings from both regions were not simultaneously obtained. Importantly, these changes coincided with the onset of spatial and habituation memory deficits, as indicated by the results from the Barnes maze and OF habituation tests. Even though no statistically significant alterations were observed in the high $\gamma$ rhythm within the hippocampus, this observation could be attributed to the inherent characteristics of high $\gamma$ activity, including its low spectral power and broad frequency range, which make it less amenable to precise analyses. These findings are in line with previous research describing alterations in hippocampal oscillations induced by oA$\beta_{1-42}$ (Sánchez-Rodríguez et al., 2017), the biologically active isoform A$\beta_{25-35}$ (Mayordomo-Cava et al., 2020) or in transgenic models (Zhang et al., 2016). Also, they support the notion that increased hippocampal $\theta$ and $\gamma$ rhythms are crucial for proper spatial memory retrieval (Vivekananda et al., 2021) because the observed reductions in these rhythms may underlie the memory impairments described in this work. Interestingly, the early decline in $\theta$ and $\gamma$ rhythms within the hippocampus was temporarily followed, although, in another cohort of animals, by an increase in the spectral power of these rhythms within the PPC. This phenomenon may suggest the existence of a compensatory mechanism that, paradoxically, might contribute to an abnormal state. This raises intriguing questions regarding the potential role of the PPC in the long-term alterations we observed in our amyloidosis model.

Furthermore, it is important to recognise the intricate relationship between neural network activity and synaptic plasticity, with each mutually influencing the other (Bikbaev & Manahan-Vaughan, 2008). Neuronal plasticity refers to the ability of the brain to adapt and remodel synapses, circuits, and neural networks in response to pathological conditions or processes that could potentially compromise their functionality or structural integrity. This adaptability leads to the generation of compensatory mechanisms, ultimately preventing the development of neuropathological disorders. Therefore, loss of synaptic plasticity may increase the vulnerability to network dysfunction or circuit loss because neural circuits and networks become less capable of adapting to pathological alterations (Contreras et al., 2023; Palop & Mucke, 2016). This raises the question of whether, in our model, hippocampal synaptic plasticity is also compromised. To address this point, we obtained hippocampal slices from mice with early amyloidosis and induced an LTP at the CA3-CA1 synapse *ex vivo*. Here, we observed that, although the HFS protocol induced LTP in vehicle-injected animals, it resulted in long-term depression (LTD) in $oA\beta_{1-42}$-treated animals. This effect was persistent even 12 days after a single $oA\beta_{1-42}$ injection. As discussed in our previous work, this modification of the induction threshold for LTP/LTD is probably attributed to the hyperexcitability induced by the peptide (Jiménez-Herrera et al., 2023; Sánchez-Rodríguez et al., 2020). This is consistent with the proposal of the Bienenstock, Cooper and Munro (BCM) theory, which suggests that a state of neuronal hyperactivity can raise the threshold for LTP induction, favouring LTD and preventing excessive neuronal activity (Cooper & Bear, 2012; Keck et al., 2017). Given the established association between neuronal oscillations and LTP (Bikbaev & Manahan-Vaughan, 2008), it is plausible that the i.c.v. administration of the $oA\beta_{1-42}$ peptide exerts an influence on long-term plasticity processes through its impact on oscillatory activity, and vice versa, potentially impacting memory and learning abilities.

## Conclusions

In conclusion, our findings offer valuable insights into the impact of early amyloidosis on neuronal oscillations and cognitive processes. Beyond the initial disruptions in hippocampal synaptic and neural network functioning, which could underlie hippocampal-dependent memory deficits, we have uncovered atypical patterns in the oscillatory activity of the PPC, a region intricately connected with the hippocampus (Ciaramelli et al., 2020). Significantly, these alterations manifest several days following hippocampal disturbances and may account for the lasting effects on spatial memory observed in our amyloidosis model. This underscores the importance of investigating other brain regions interconnected with the hippocampus to fully comprehend the spatiotemporal progression of early amyloid pathology and suggests potential future research into therapeutic strategies aimed at addressing these initial neural network disruptions in AD. Moreover, given the suitability of EEG metrics as potential biomarkers for the preclinical stages of AD (Gaubert et al., 2019) and the association of PPC dysfunction with a higher risk of MCI to AD conversion (Ilardi et al., 2022), our findings could prompt further studies exploring PPC oscillatory activity as a viable method for early AD detection and intervention.

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

## Additional information

### Data availability statement

All data supporting the results in this study are provided in the final publication itself and the Supporting information. Raw data and recordings are available in the laboratory from the corresponding authors and will be provided upon reasonable request.

### Competing interests

The authors declare that they have no competing interests.

### Author contributions

L.J.D. and J.D.N.L. were responsible for the initial conceptualisation. R.S.C. designed the optimised analytical-experimental approach. R.J.H., A.C., G.I.L. and S.D. performed the experiments. V.C.A. and R.S.C. contributed with new analytical/computational tools. R.J.H., A.C., G.I.L., S.D., V.C.A. and R.S.C. analysed the data. S.D., A.C., V.C.A., R.S.C., L.J.D. and J.D.N.L. were responsible for visualisation. S.D. and A.C. were responsible for writing the original draft. S.D., A.C., R.S.C., J.D.N.L. and L.J.D. performed the reviewing and editing. L.J.D., J.D.N.L. and R.S.C. were responsible for funding acquisition. L.J.D., J.D.N.L. and R.S.C. were responsible for supervision and project administration. All authors read and approved the final version of the manuscript submitted for publication.

### Funding

This work was supported by the MCIN/AEI/10.13039/501100011033 (grant numbers PID2020-115823-GBI00; PID2024-155413NB-I00), JCCM/ERDF – a way of making Europe (grant numbers SBPLY/21/180501/000150; SBPLY/24/180225/000181) and UCLM/ERDF (2022-GRIN-34354; 2025-GRIN-38530 grants) to JDNL and LJD, as well as ANDALUSIA/ERDF (UPO-1380660) and MCIN/AEI (PID2022-141997NB-I00) both to RSC. AC held a Margarita Salas Postdoctoral Research Fellow (2021-MS-20549) funded by the European Union NextGenerationEU/PRTR. GIL held a predoctoral fellowship granted by the UCLM/ESF Plan Propio de Investigación Programme.

### Keywords

Alzheimer's disease, amyloid-$\beta$ (A$\beta$), hippocampus, LTP, oligomers, oscillatory activity, posterior parietal cortex, spatial memory

## Supporting information

Additional supporting information can be found online in the Supporting Information section at the end of the HTML view of the article. Supporting information files available:

**Peer Review History**

