## [Peer Review History · The Journal of Physiology]

Posterior parietal cortex oscillatory activity reflects persistent spatial memory impairments induced by early hippocampal amyloidosis in male mice

Souhail Djebari, Ana Contreras, Victor D. Castro-Andres, Raquel Jimenez-Herrera, Guillermo Iborra-Lazaro, Raudel Sanchez-Campusano, Lydia Jimenez-Diaz, and Juan D. Navarro López
DOI: 10.1113/JP286196

Corresponding author(s): Juan Navarro López (juan.navarro@uclm.es)

The following individual(s) involved in review of this submission have agreed to reveal their identity: Sean Williams (Referee #3)

Review Timeline:

Submission Date:	25-Jun-2024
Editorial Decision:	24-Jul-2024
Revision Received:	07-Jul-2025
Editorial Decision:	22-Sep-2025
Revision Received:	27-Oct-2025
Editorial Decision:	03-Dec-2025
Revision Received:	16-Jan-2026
Accepted:	09-Feb-2026

Senior Editor: Nathan Schoppa

Reviewing Editor: Valentina Mosienko

Transaction Report:

Dear Dr Navarro López,

Re: JP-RP-2024-286196 "Posterior parietal cortex oscillatory activity reflects persistent spatial memory impairments induced by early hippocampal amyloidosis in male mice" by Souhail Djebari, Ana Contreras, Raquel Jimenez-Herrera, Victor D. Castro-Andres, Guillermo Iborra-Lazaro, Raudel Sanchez-Campusano, Lydia Jimenez-Diaz, and Juan D. Navarro López

Thank you for submitting your manuscript to The Journal of Physiology. It has been assessed by a Reviewing Editor and by 2 expert referees and we are pleased to tell you that it is potentially acceptable for publication following satisfactory major revision.

LANGUAGE EDITING AND SUPPORT FOR PUBLICATION: If you would like help with English language editing, or other article preparation support, Wiley Editing Services offers expert help, including English Language Editing, as well as translation, manuscript formatting, and figure formatting at www.wileyauthors.com/eoo/preparation. You can also find resources for Preparing Your Article for general guidance about writing and preparing your manuscript at www.wileyauthors.com/eoo/prepresources.

REVISION CHECKLIST:

We look forward to receiving your revised submission.

Yours sincerely,

David Wyllie
Senior Editor
The Journal of Physiology

REQUIRED ITEMS FOR REVISION

- Include a Key Points list in the article itself, before the Abstract.
- Author photo and profile. First or joint first authors are asked to provide a short biography (no more than 100 words for one author or 150 words in total for joint first authors) and a portrait photograph. These should be uploaded and clearly labelled together in a Word document with the revised version of the manuscript. See Information for Authors for further details.

- Please upload separate high-quality figure files via the submission form.

- Papers must comply with the Statistics Policy: https://jp.msubmit.net/cgi-bin/main.plex?form_type=display_requirements#statistics.

In summary:

- If n {less than or equal to} 30, all data points must be plotted in the figure in a way that reveals their range and distribution. A bar graph with data points overlaid, a box and whisker plot or a violin plot (preferably with data points included) are acceptable formats.
- If $n > 30$, then the entire raw dataset must be made available either as supporting information, or hosted on a not-for-profit repository, e.g. FigShare, with access details provided in the manuscript.
- 'n' clearly defined (e.g. x cells from y slices in z animals) in the Methods. Authors should be mindful of pseudoreplication.
- All relevant 'n' values must be clearly stated in the main text, figures and tables.
- The most appropriate summary statistic (e.g. mean or median and standard deviation) must be used. Standard Error of the Mean (SEM) alone is not permitted.
- Exact p values must be stated. Authors must not use 'greater than' or 'less than'. Exact p values must be stated to three significant figures even when 'no statistical significance' is claimed.

- Please include an Abstract Figure file, as well as the Figure Legend text within the main article file. The Abstract Figure is a piece of artwork designed to give readers an immediate understanding of the research and should summarise the main conclusions. If possible, the image should be easily 'readable' from left to right or top to bottom. It should show the physiological relevance of the manuscript so readers can assess the importance and content of its findings. Abstract Figures should not merely recapitulate other figures in the manuscript. Please try to keep the diagram as simple as possible and without superfluous information that may distract from the main conclusion(s). Abstract Figures must be provided by authors no later than the revised manuscript stage and should be uploaded as a separate file during online submission labelled as File Type 'Abstract Figure'. Please also ensure that you include the figure legend in the main article file. All Abstract Figures

should be created using BioRender. Authors should use The Journal's premium BioRender account to export high-resolution images. Details on how to use and access the premium account are included as part of this email.

EDITOR COMMENTS

Reviewing Editor: Ethics Concerns:

Please make sure that methods of euthanasia for all experiments is stated - currently details on methods of euthanasia for animals that were used for IHC/staining analysis are missing.

Comments for Authors to ensure the paper complies with the Statistics Policy:

Please make sure that the paper complies with the statistical policy of the Journal - specifically, all data points should be displayed on figures, and SD rather than SEM should be provided; n number should be clearly stated in figure legends.

Comments to the Author:

The reviewers highlighted a number of issues that should be addressed as part of the revision. In particular, Reviewer 1 raised significant points regarding the statistical analysis - please make sure to thoroughly address those in the revised version of the manuscript.

The Journal does not encourage including supporting information as part of the submission - please include the data if possible in the main part of the manuscript if they are essential for understanding.

Please make sure that all graphs comply with the statistical policy of the journal.

Senior Editor:

Comments for Authors to ensure the paper complies with the Statistics Policy:

Please ensure that any revision conforms to your Statistics Policy in terms of data reporting

Comments to the Author:

Your manuscript has been assessed by two expert referees and based on their comments the work is not suitable for further consideration in its present form. However, each referee does comment positively on aspects of this work but with the caveat that the statistical analysis needs to be re-done as there are several queries regarding the claims made. Clearly, re-analyzing may result in a different interpretation of your findings which would of course likely reduce the impact of your study. Thus, I offer the opportunity to thoroughly revise to address the comments raised but caution that should the interpretation of your results yield different conclusions then this would likely influence our decision to proceed further with this work.

REFEREE COMMENTS

Referee #1:

Djebari and colleagues investigated the role of the posterior parietal cortex (PPC) and the hippocampus (HC) in amyloidosis induced memory dysfunction. In addition to a number of memory tests, they recorded LFP in the PPC and the HC, and assessed LTP in the HC. They report impaired memory, changes in oscillatory PPC and HC activity, and reduced LTP in HC slices. Overall, experimental design and behavioral controls are well done. My only major concern are the statistical analyses, which seem wrong or opaque in some places. Overall, I suggest that the authors get a statistician to check their analyses and perhaps redo using mixed effects models with animals and repeated measures as random effects.

Major:

1. Figure 3A: The measure of spectral power is unclear. Some papers report an underlying $1/f$ power spectrum with a maximal power around 1 Hz also for rat HC, some don't. Is there an explanation why this differs between studies (filter?) Is percentage based on power or log-power? D-1 and D-3 were recorded during Barnes maze training, D-12 during test. What was the condition during PRE?

Since vehicle and oAb1-42 are directly compared, it would make more sense to show those in the same figure, not in two distinct figures. Also, there is no indication of variance in this figure. It would be beneficial to see the individual animals' curves. That could help elucidate whether the "bumps" in the oAb1-42 group at 25 Hz and 55 Hz are coming from individual animals or are a general feature of this group.

2. Statistical analyses seem wrong. For a group comparison with ~6 animals, I would not expect $F(1,1432)$ degrees of freedom. I have the impression that repeated measures were taken as independent measurement in these analyses. The varying degrees of freedom apply to most analyses. Similarly, the small SEM indicated in Fig. 3 seem to be based on trials instead of animals. This explains why p values are so small although the configuration of data points in the figure seem quite variable.

3. Figure 4A/D: Why are results presented as this rather complicated "probability density" instead of average theta power? Figure 4B/C: the figures cannot represent power "in % of PRE" since PRE is included and not 100%.

4. On L550ff, a significant difference for control animals was reported, but no significant effect was found in treated mice. Any conclusion regarding differences between the two groups requires to be based on a significant interaction group x decrease.

5. Figures 5B/D: There are variances indicated for the OF1/PRE groups although those are set to 100%. If all measures have been set to 100%, they cannot have any variance.

6. L1327ff: Here the authors claim that theta-beta interactions on day PRE are likely attributable to oAb-induced changes. How can this be if oAb was administered only after PRE?

Minor:

1. Figure 2: Legend says it shows oscillatory activity which it doesn't.

2. Methods and L479: I was missing more detail on the fourier transforms in the methods. Some of that can be found in L479ff. Please extend the description of FFT/mTFT parameters in the methods.

3. Please provide sample sizes for every analysis. Statements like "Only those animals that provided a clear and uninterrupted 5-min segment of recording were considered" are not sufficient.

4. L1356: CI: confidence interval (not confidential)

Referee #2:

In this study, Djebari et al investigate the consequences of induced hippocampal amyloidosis (as a mouse model of AD) both at a behavioural and neurophysiological level. They demonstrate persistent oscillatory changes, in particular within theta frequency, in both hippocampus and posterior parietal cortex (PPC) associated with affected spatial learning. This is an interesting finding in isolation, and experiments are well thought through. However the choice to focus on PPC may not be as translatable to humans where other regions have been critically associated with spatial memory e.g. striatum, retrosplenium. In addition, the inability to simultaneously record from the hippocampus and PPC limits conclusions that can be made about their interaction. In addition, there are in my opinion some absent information and methodological queries

that I will outline below

1. Is there any data to quantify variable Abeta expression in the 'disease' mouse models that can be related to behavioural outcome
2. It appears from figure 2 that learning differences between vehicle and disease animals narrows over time (e.g. Day 12) - do the authors feel effect on spatial memory is persistent?
3. It appears a broadband frequency increase in power was observed in the disease animals (Figure 3) which is surprising. Do the authors feel this reflects a spectral shift phenomenon? As I understand each LFP recording was normalised to pre-treatment which may introduce bias versus correcting across the mean/z score of all recordings.
4. Subsequent sliding window analysis (line 482) highlighted theta activity as most important factor in spectral change. This should be shown with appropriate statistics
5. As I understand LFP recording was performed after task performance not during, this should be acknowledged as a limitation. In addition can the authors confirm animals were awake and alert during this period
6. The authors found that theta and low gamma power were decreased within the hippocampus in the disease animals, was any theta-gamma coupling analysis attempted. In addition, was repeated longer term assessment (e.g. days later) attempted as with the PPC experiments.
7. Again relation between amount of Abeta expression within the hippocampus and LTP result (figure 5) would be important to understand

END OF COMMENTS

POINT-BY-POINT REPLY TO THE REVIEWS

We greatly appreciate the opportunity to revise and resubmit our manuscript. We would also like to thank the Reviewers for their detailed and constructive comments, which have undoubtedly contributed to improving the manuscript.

In response to the reviewers' suggestions, new analyses have been conducted and figures reorganized. All figures now display standard deviation rather than standard error. To better understand the spectral properties of the LFP signals recorded in the PPC, we conducted a detailed analysis of their periodic and aperiodic components included now in the new Figure 2. Figure 3, which illustrates alterations in PPC oscillatory activity following $\alpha\beta$ intracerebroventricular injection, has been substantially redesigned and now provides a more comprehensive representation of spectral power changes across the experimental sessions. Moreover, Figure 6 now reports the spectral analysis across different frequency bands, covering both intra-condition and inter-band comparisons.

We have also clarified several points of confusion that were likely due to methodological limitations or insufficient explanations, including the relationship between electrophysiological and behavioral findings. All changes have been highlighted in red in the revised manuscript.

We sincerely thank the Reviewers and Editors for their valuable comments and thoughtful insights, which have significantly enhanced the clarity and overall quality of the manuscript. We hope that the revised version now meets the standards of *The Journal of Physiology*.

We believe that we have addressed all concerns raised by the Reviewers and that the revised manuscript now presents our original conclusions in a clearer and more robust manner.

EDITOR COMMENTS

Reviewing Editor: Ethics Concerns:

Please make sure that methods of euthanasia for all experiments is stated - currently details on methods of euthanasia for animals that were used for IHC/staining analysis are missing.

Comments for Authors to ensure the paper complies with the Statistics Policy:

Please make sure that the paper complies with the statistical policy of the Journal - specifically, all data points should be displayed on figures, and SD rather than SEM should be provided; n number should be clearly stated in figure legends.

Comments to the Author:

The reviewers highlighted a number of issues that should be addressed as part of the revision. In particular, Reviewer 1 raised significant points regarding the statistical analysis - please make sure to thoroughly address those in the revised version of the manuscript.

The Journal does not encourage including supporting information as part of the submission - please include the data if possible in the main part of the manuscript if they are essential for understanding.

Please make sure that all graphs comply with the statistical policy of the journal.

Senior Editor:

Comments for Authors to ensure the paper complies with the Statistics Policy:

Please ensure that any revision conforms to your Statistics Policy in terms of data reporting

Comments to the Author:

Your manuscript has been assessed by two expert referees and based on their comments the work is not suitable for further consideration in its present form. However, each referee does comment positively on aspects of this work but with the caveat that the statistical analysis needs to be re-done as there are several queries regarding the claims be made. Clearly, re-analyzing may result in a different interpretation of your findings which would of course likely reduce the impact of your study. Thus, I offer the opportunity to thoroughly revise to address the comments raised but caution that should the interpretation of your results yield different conclusions then this would likely influence our decision to proceed further with this work.

REFEREE COMMENTS

Referee #1:

Djebari and colleagues investigated the role of the posterior parietal cortex (PPC) and the hippocampus (HC) in amyloidosis induced memory dysfunction. In addition to a number of memory tests, they recorded LFP in the PPC and the HC, and assessed LTP in the HC. They report impaired memory, changes in oscillatory PPC and HC activity, and reduced LTP in HC slices. Overall, experimental design and behavioral controls are well done. My only major concern are the statistical analyses, which seem wrong or opaque in some places. Overall, I suggest that the authors get a statistician to check their analyses and perhaps redo using mixed effects models with animals and repeated measures as random effects.

Major:

1. Figure 3A: The measure of spectral power is unclear. Some papers report an underlying 1/f power spectrum with a maximal power around 1 Hz also for rat HC, some don't. Is there an explanation why this differs between studies (filter?) Is percentage based on power or log-power?

We sincerely thank the reviewer for raising this key point, which prompted us to reanalyze our spectral data in depth. We acknowledge that in the original version of the manuscript, the rationale behind the normalization of power spectra and the representation of Figure 3A was not sufficiently detailed, and we take full responsibility for that.

Regarding previous Figure 3A, the aim was to display the spectral power of the different frequency bands to illustrate each band's contribution to the oscillatory activity in the parietal cortex. To achieve this, the power of each frequency was normalized relative to the maximum power in each group (control or $\alpha\text{A}\beta_{1-42}$), which in both cases corresponded to ≈ 8 Hz, thus within the *theta* rhythm the fundamental component centered at ≈ 8 Hz in the frequency band (θ ; 4-12 Hz). This was done to highlight that, although the frequencies are analyzed separately, they all do not contribute equally to the animals' oscillatory activity, with the *theta* rhythm being the most dominant.

However, the comments of the reviewer prompted us to perform a deeper analysis of spectral power, allowing us to represent it in a more understandable manner. We have modified the figure to display the normalization of each raw spectrum with respect to the total power (in percentage, %) for improved clarity (**new Figure 4A-B**, see below). The Results section (page 18, lines 508-514) and figure legend (pages 20-21 and 50-51, lines 549-554 and 1408-1414) have also been modified accordingly. Additionally, we have included an alternative version of the figure (available for Referee #1's consideration) displaying the raw spectral power in mV^2 , should the reviewer prefer this format, allowing a comparison focused on absolute power differences between groups, rather than the relative distribution of power across frequencies, as shown in the original % total power representation. Note that, in the two representations, the contribution of the *theta* (θ ; 4-12 Hz) rhythm centered on the fundamental harmonic of ≈ 8 Hz was very predominant compared to the contributions to the power spectrum of the other adjacent frequency bands [*delta* (δ ; 0-4 Hz), *beta* (β ; 12-30 Hz) or low gamma (γ ; 30-48 Hz)], which means that the power relation $P(f) = 1/f$ (or law of slope equal to -1 in the log-log plot) does not hold, at least for the spectral powers in the frequency range from 1 to 48 Hz. We have shown that a general power function of type $P(f) = C * f^\alpha$ was fulfilled the spectra of the LFP recorded from PPC, where C determines the reference power value when the frequency is 1 Hz (or intercept) and α determines an alternative power law (or slope, $\alpha \neq -1$).

Both the intercept and slope are obtained from the regression line [applying logarithm; $\log_{10} P(f) = \log_{10} C + \alpha \log_{10} f$] between the powers and frequencies in the log-log representation. A new figure showing all the steps in the procedure has been included in the Methods section for clarity (**new Figure 2**). Regression lines in the log-log plot for the frequency range 1–48 Hz showed moderate values of the determination coefficients ($0.65 < R^2 < 0.85$), indicating a limited strength of the fit between power vs. frequency points, with slopes much greater than -1 (range [-0.56, -0.33]; mean = -0.46). On average, the point cloud was best fitted to a power-frequency relationship of the type $P(f) = C/\sqrt{f}$ by slope approximation ($\alpha \approx -0.5$); see **Figure 2D-L**. This means that the aperiodic components of the spectra should not be considered for the spectral analysis of the recorded LFPs, and this is the key reason why in this study we focused on the periodic components, mainly on the fundamental harmonic in the *theta* (θ ; 4-12 Hz) band. Therefore, for a more detailed analysis of the periodic components of the spectrum, we thoroughly studied the time-frequency distribution of the spectral powers (spectrogram) by using multi-Taper Fourier Transform (mTFT), a more robust technique than conventional fast Fourier transform (FFT).

On the other hand, we point out here that we have not applied any filter that affects the 1 Hz component in the *delta* band (see **Figure 2A**). On the contrary, our band-pass FIR filter was designed in the frequency range of 0.1 to 150 Hz, and our band-stop IIR filter only removed the 50 Hz interference noise (from 48 to 52 Hz). Note that when transforming the data to a logarithmic scale (**Figure 2B-C**) on the frequency axis, values less than 1Hz cannot be observed because in that range the logarithmic function is undetermined. For this reason, the *delta* frequency band is defined from 1 to 4 Hz.

Finally, as shown in the **new Figure 4A-B** and in the **Figure 2A-C**, we did not observe a maximum power around 1 Hz in our spectra, neither for the LFPs recorded in HC nor for those recorded in PPC, on the contrary, the maximum power of all our spectra was in the *theta* (θ ; 4-12 Hz) band centered at 8 Hz.

Figure 2. Different representations of the mean power spectra of the LFPs recorded from PPC of the mice in the two groups (Vehicle and $\alpha\beta_{1-42}$) across the days (PRE, D-1, D-3, D-12). (A) Linear-Linear plot of the spectra showing the spectral component in the range 0-2 Hz with peak at 1 Hz (see vertical black arrows, < 2 Hz) within the *delta* (δ ; 0-4 Hz) band. **(B)** Linear-Log plot of the spectra where the x-axis starts at 1 Hz because the base-10 logarithmic function of power is undefined for small values (< 1 Hz) of frequency. This is the representation adopted for this study because it focuses on the fundamental periodic component (≈ 8 Hz) in the *theta*

(θ ; 4-12 Hz) frequency band. **(C)** Log-Log plot of the spectra with low dispersion (intercepts in the range [2.86, 5.68]) that makes it difficult to differentiate the spectra. **(D)** Log-Log plot of the aperiodic components of the LFP spectra for the two range of frequencies (range 1: 1-48 Hz; range 2: 52-150 Hz). Note that power-frequency relation $P(f) = 1/f$ (or law of slope equal to -1 in the log-log plot) does not hold in the frequency range of 1-48 Hz (slopes in the range [-0.56, -0.33]; mean = -0.46). On average, the point cloud was best fitted to a power-frequency relationship of the type $P(f) = C/\sqrt{f}$ by slope approximation ($\alpha \approx -0.5$). **(E-F)** Each panel corresponds to the linear fit between power vs. frequency points in each experimental condition. Linear equations ($Y = slope * X + Intercept$) and the corresponding determination coefficients (R^2) are indicated. All representations correspond to power spectra normalized with respect to the total power (see y-axis). Plots in A-C show the notch (see red arrows) resulting from filtering the 50 Hz interference noise, with cutoff frequencies at 48 Hz and 52 Hz.

Alternative Figure 4 to replace new Figure 4 if required by Reviewer 1. Panel shows raw spectral power in mV² instead of the percentage of total power.

D-1 and D-3 were recorded during Barnes maze training, D-12 during test. What was the condition during PRE?

We regret any confusion that our experimental design may have caused. LFP recordings were performed right after the behavioral task in D-1, D-3 and D-12, in a registration box in alert, freely moving mice, as the nature of the behavioral tests we conducted does not allow for simultaneous *in vivo* recordings. During PRE-treatment day, LFPs were recorded in that same box, and approximately at the same time of day to avoid influences of circadian rhythms. The necessity for animals to freely enter an escape hole in the Barnes maze task precludes the possibility of attaching wires to their heads, as this would inhibit their movement and ability to complete the task. Similarly, the OF task was performed in a the LABORAS® system, a platform that converts mechanical vibrations from animal movements into electrical signals. The sensitivity of this system means that any additional vibrations, such as those from a wire, could distort the results, undermining the integrity of the data collected. Also, LABORAS® system requires calibration before each session, including the animal's weight as a parameter. Thus, introducing any external element into the cage, like wires, could affect the accuracy of these measurements. This aspect has been clarified in the Method section of the ms. (Page 9, lines 227-228).

Since vehicle and oAb1-42 are directly compared, it would make more sense to show those in the same figure, not in two distinct figures. Also, there is no indication of variance in this figure. It would be beneficial to see the individual animals' curves. That could help elucidate whether the "bumps" in the oAb1-42 group at 25 Hz and 55 Hz are coming from individual animals or are a general feature of this group.

We agree with the referee that displaying the curves for both groups in the same graph in Figure 3A would facilitate comparison. However, we opted to present them in separate graphs to avoid excessive overlap, as multiple curves would otherwise superimpose and hinder a clear visualization of the results. The aim of this figure was to display the spectral power of the different frequency bands to illustrate each band's contribution to the oscillatory activity in the parietal cortex. To achieve this, the power of each frequency was normalized relative to the maximum frequency power in each group (control or oAb₁₋₄₂), which in both cases corresponded to ≈ 8 Hz, thus within the *theta* rhythm—the fundamental rhythm centered at ≈ 8 Hz in the *theta* frequency band (4-12 Hz). This was done to highlight that, although the frequencies are analyzed separately, they do not all contribute equally to the animals' oscillatory activity, with the theta rhythm being the most dominant.

However, at no point did we intend to make a direct comparison between the control and oAb₁₋₄₂ groups. To achieve this, we performed a separate comparison for each frequency (new Figure 4C-H), normalizing with respect to the pre-injection session (when all animals were non-injected naïve) for each group independently. This approach allows us to track how the treatment affects the oscillatory activity within each group compared to their pre-treatment baseline, ensuring a longitudinal assessment of changes. This method also maintains consistency with the other tests in the study (new Fig. 7), thereby facilitating the reader's understanding.

2. Statistical analyses seem wrong. For a group comparison with ~6 animals, I would not expect F(1,1432) degrees of freedom. I have the impression that repeated measures were taken as independent measurement in these analyses. The varying degrees of freedom apply to most analyses. Similarly, the small SEM indicated in Fig. 3 seem to be based on trials instead of animals. This explains why p values are so small although the configuration of data points in the figure seem quite variable.

We appreciate the referee's observation and would like to clarify this point. After a comprehensive review of our data and leveraging our prior experience, we established a 5-min window as the optimal duration to achieve clean, consistent recordings across all experimental subjects. By means of fast Fourier transform (FFT), the average power spectra were calculated from 10-s epochs (18 trials) covering the 3-min analysis window of recordings acquired from 10 mice per group ($N_E = 180$ epochs). This windowing procedure using FFT is statistically more robust than analyzing the entire 3 min directly. The advantage of using more trials includes reduced bias in the spectrum estimate, and these trials can be assumed to be interchangeable, which is in line with the permutation test requirements (i.e., the sequence of selected trials throughout the entire epoch does not influence the statistical significance of the results).

However, following the comments of the reviewer, we re-examined and reanalyzed the recordings using revised criteria, which allowed us to include the recordings of additional animals from cohort 1 and thus increase our sample size. As shown in the **updated Figure 4D-H**, the statistical analysis of the differences among the mean values of the spectral power from the two groups of mice ($\alpha\beta_{1-42}$ and vehicle; $n = 10$ mice per group) across the four experimental conditions (days: PRE, D-1, D-3, D-12) has been performed considering only 10 independent measurements (one for each mouse) with 18 different observations (18 trials of 10 s each). In this way, the degrees of freedom were obtained as: $df1 = \# \text{ of groups} - 1 = 1$ and $df2 = (\# \text{ of independent measurements} - 1) \times (\# \text{ of total experimental conditions}) = 72$. In the **new Figure 4D-H** and in the Results section (page 19, lines 521-546) all statistical reports have been corrected.

3. Figure 4A/D: Why are results presented as this rather complicated "probability density" instead of average theta power? Figure 4B/C: the figures cannot represent power "in % of PRE" since PRE is included and not 100%.

We thank the reviewer for raising this interesting point. The time-frequency localization of brain rhythms and the probability density analysis are two key steps for understanding spectral patterns. In this study the added value of spectrograms of the Figure 4B-C (now figure 5B-C) is justified both methodologically and physiologically. Methodologically, the time-frequency decomposition of LFP recording has shown us that the windowing procedure carried out to calculate the mTFT (6 epochs of 300-s duration each, with the epochs being interchangeable within each of the 8 experimental conditions) was feasible in the statistical sense. Note that the theta fundamental oscillation (centered at 8 Hz) was significant throughout the entire LFP recording, which ensures that it is not a seasonal (i.e., always located in an isolated segment of the recording) spectral pattern, nor is it a random spectral event (i.e., randomly located in some segments of the recording). Physiologically, the time-frequency decomposition of the spectral power using mTFT has shown us the poor occurrence of secondary harmonics (i.e., delta, beta and gamma oscillations of moderate amplitude and very little presence in the time domain) along the neurophysiological time series, in contrast with the sustained occurrence of a theta fundamental harmonic in the LFP from PPC with a potential role in the long-term alterations we observed in our amyloidosis model, an aspect that cannot be appreciated from the conventional (power vs. frequency) analysis with FFT. In accordance with this spectral analysis, the histograms of probability density (new Figure 5A and D) for selected frequency bands were obtained based on Jackknifed estimation of variance between pairs of spectrograms (time-frequency plots of the LFP powers), covering all frequency values (resolution, $df = 0.5$ Hz) within each time window (resolution, $dt = 0.5$ s). We agree with the reviewer that this analysis is more complicated but is certainly essential for the statistics of multiple comparisons between power spectrograms, a procedure slightly different, but more robust than the conventional peak-mean analysis of spectral power calculated by FFT. For

more details, the reviewer can refer to the following papers published by one of us (R.S.-C.) using these analyses.

Lintas A, **Sánchez-Campusano R**, Villa AEP, Gruart A & Delgado-García JM. (2021). Operant conditioning deficits and modified local field potential activities in parvalbumin-deficient mice. *Sci Rep* **11**, 2970.

Reus-García MM, **Sánchez-Campusano R**, Ledderose J, Dogbevia GK, Treviño M, Hasan MT, Gruart A & Delgado-García JM. (2021). The claustrum is involved in cognitive processes related to the classical conditioning of eyelid responses in behaving rabbits. *Cerebral Cortex* **31**, 281-300.

Conde-Moro AR, Rocha-Almeida F, **Sánchez-Campusano R**, Delgado-García JM & Gruart A. (2019). The activity of the prelimbic cortex in rats is enhanced during the cooperative acquisition of an instrumental learning task. *Prog Neurobiol* **183**, 101692.

Fernández-Lamo I, **Sánchez-Campusano R**, Gruart A & Delgado-García JM. (2016). Functional states of rat cortical circuits during the unpredictable availability of a reward-related cue. *Sci Rep* **6**, 37650.

4. On L550ff, a significant difference for control animals was reported, but no significant effect was found in treated mice. Any conclusion regarding differences between the two groups requires to be based on a significant interaction group x decrease.

We thank the reviewer for his/her observation. Indeed, when evaluating habituation memory within each group, we used a paired-samples t-test, as this provides more precise insight into the evolution of each group of animals across the two sessions. However, we agree that a two-way ANOVA would be more appropriate for assessing treatment effects between groups.

Unfortunately, with the initial sample size, although a clear trend was observed, it was insufficient to reach statistical significance with that analysis. Therefore, although this test is already well-characterized in our laboratory using this model, where the impact of hippocampal amyloidosis on habituation memory has been clearly demonstrated (Sánchez-Rodríguez *et al.*, 2020; Jiménez-Herrera *et al.*, 2023; Djebari *et al.*, 2025), we decided to include a small additional subset of animals to increase the sample size and reinforce the robustness of these results.

As a result, the number of animals per group is now 11, and significant treatment-related differences were observed. This information has been incorporated into the Results section (page 25, lines 680-692), and Figures 1 and 7 have been updated accordingly to reflect the expanded dataset.

5. Figures 5B/D: There are variances indicated for the OF1/PRE groups although those are set to 100%. If all measures have been set to 100%, they cannot have any variance.

We appreciate the referee's comment and believe there may have been a misunderstanding. First, we would like to clarify that Figure 5B (now figure 7B) presents two significant differences. The # symbol indicates a significant decrease in the distance traveled by vehicle-treated animals between OF1 and OF2, reflecting habituation (an effect observed only in the vehicle group, not in the $\alpha\beta_{1-42}$ -treated animals). In contrast, the * symbol denotes a significant difference between the vehicle and $\alpha\beta_{1-42}$ groups in OF2. This aligns with the

previous observation, as vehicle-treated animals reduced their distance traveled in OF2, whereas oA β_{1-42} -treated animals did not.

Additionally, we would like to highlight that in Figure 7B, the mean value for OF1 has been set to 100%. The distance traveled by each animal in OF1 was averaged relative to this reference point, which accounts for the variance mentioned by the referee. We believe this approach is important, as it demonstrates the low variability in distance traveled by different animals during OF1, thereby strengthening the validity of our results. A similar normalization was applied to the PRE condition in Figures 7D–H, where the mean was set to 100%, and each animal's value was averaged accordingly to reflect variance on that recording day.

6. L1327ff: Here the authors claim that theta-beta interactions on day PRE are likely attributable to oAb-induced changes. How can this be if oAb was administered only after PRE?

We greatly appreciate the referee's observation, which helped us identify an error in the presentation of our results. As the treatment had not yet been administered, the difference in the PRE condition could indeed not be attributed to its effects. We have now corrected this section (now included in Result section (page 22, lines 603-616) and new Figure 6) accordingly removing that statement.

Minor:

1. Figure 2: Legend says it shows oscillatory activity which it doesn't.

We appreciate the referee's attention to this error, which resulted from a previous version of the figure. The figure legend has been corrected accordingly in the revised ms. (new Figure 3; page 18 and 49, line 493 and 1398).

2. Methods and L479: I was missing more detail on the fourier transforms in the methods. Some of that can be found in L479ff. Please extend the description of FFT/mTFT parameters in the methods.

As described in the previous point, averaged power spectra were calculated from 10-s epochs (18 trials) covering 3 minutes of the recordings acquired from 10 mice per group ($N_E = 180$ epochs). This windowing procedure using FFT (**new Figure 4A-B**) is statistically more efficient than the mere spectral analysis from entire 3-minutes epoch because more trials (or epochs) lead to lower bias in spectrum estimate and these trials can be assumed to be interchangeable, which is in line with the permutation test requirements (i.e., that the order of the selected trials throughout the entire epoch does not affect the statistical significance of the results). Using FFT, it was possible to obtain both the raw spectra and the normalized power spectra with respect to the total power. By normalizing, the effect of absolute power differences between the groups is removed, allowing the comparison to focus on the relative distribution of power across different frequencies. (These aspects were included in the main text under the section "In vivo electrophysiological recordings and analysis" for clarity).

On the other hand, spectrograms shown in Figure 4 (now figure 5) were calculated by multi-Taper Fourier Transform (mTFT). For a selected time, resolution of $dt = 0.5$ s, the resulting frequency resolution of the spectrograms was $df = 0.5$ Hz, in accordance with the Fourier uncertainty principle: $\Delta t \times \Delta f \geq \text{Constant}$. This principle establishes that the alteration in the values of one of the two simultaneous measurements Δt and Δf in a specific state produces

the updating of the values of the other through its measurement, and therefore, very poor temporal resolution leads to a very precise frequency resolution, and vice versa. The appropriate thing is to find optimal values for both magnitudes without violating the uncertainty principle, for instance: spectrograms with time-steps of 0.5 s covering 300 s and frequency-steps of 0.5 Hz covering 150 Hz.

The mTFT method uses a family of Slepian sequences called tapers. These sequences have the property that for a given short time-window of data ($T = 2$ s) and a narrow frequency-bandwidth ($W = 1$ Hz), the first K tapers ($K = 2TW - 1$, number of tapers $K = 3$) are optimally concentrated in the frequency range $[f - W, f + W]$ and the multi-tapers methods provides estimates with good bias-variance characteristics. Therefore, since Slepian sequences are mutually orthogonal and the trials are interchangeable, the spectrum estimates computed from the different tapers and trials are statistically independent and averaging over them reduces the variance of the estimate. Thus, multi-taper estimates of the spectrogram involving K tapers and N_E epochs are based on computing $m = N_E \times K$ Fourier transforms. (This paragraph was included in the main text under the section "In vivo electrophysiological recordings and analysis" for clarity).

All this information has been added to the Methods section (pages 9-10, lines 238-264).

3. Please provide sample sizes for every analysis. Statements like "Only those animals that provided a clear and uninterrupted 5-min segment of recording were considered" are not sufficient.

We completely agree with the referee and apologize for the lack of clarity. The number of animals used (n) is specified for each experiment, and we have now included it in the sections where it was previously missing in the ms. (page 9, line 232).

4. L1356: CI: confidence interval (not confidential)

We apologize for the error and have corrected it in the revised ms. (pages 24 and 54, lines 667 and 1468).

Referee #2:

In this study, Djebari et al investigate the consequences of induced hippocampal amyloidosis (as a mouse model of AD) both at a behavioural and neurophysiological level. They demonstrate persistent oscillatory changes, in particular within theta frequency, in both hippocampus and posterior parietal cortex (PPC) associated with affected spatial learning. This is an interesting finding in isolation, and experiments are well thought through. However the choice to focus on PPC may not be as translatable to humans where other regions have been critically associated with spatial memory e.g. striatum, retrosplenium. In addition, the inability to simultaneously record from the hippocampus and PPC limits conclusions that can be made about their interaction. In addition, there are in my opinion some absent information and methodological queries that I will outline below.

1. Is there any data to quantify variable Abeta expression in the 'disease' mouse models that can be related to behavioural outcome

First, we would like to thank the reviewer for his/her insightful comments, which have significantly contributed to improving and clarifying our ms.

We agree that this is a highly relevant point and appreciate the reviewer's attention to this aspect. In fact, we have previously demonstrated that a single icv. injection of oA β_{1-42} is sufficient to induce localized acute amyloidosis, with effects that persist for at least 17 days post-injection (Jiménez-Herrera *et al.*, 2023). We also showed that this injection reaches the hippocampus, where oligomeric forms of the peptide can be detected by western blot using a specific antibody against the molecular weight band corresponding to the oligomers. These are visible as early as 1 hour post-injection and persist for at least 24 hours (see additional file 1 in (Jiménez-Herrera *et al.*, 2023); the figure has been included under these lines to facilitate its visualization by the reviewer). Here, we also demonstrated that even 17 days after a single injection, when oA β_{1-42} is no longer detectable in the hippocampus, its detrimental effects on LTP and hippocampal-dependent memory persist. This suggest that the observed alterations are not strictly dependent on the continued presence or concentration of oA β_{1-42} in the tissue. Instead, the peptide likely triggers a cascade of pathological events that lead to long-lasting dysfunction.

Additional file 1 in (Jiménez-Herrera *et al.*, 2023)

Additionally, we previously conducted a histological study of $\text{oA}\beta_{1-42}$ deposition along the rostrocaudal axis of the hippocampus (see supplementary file 1 in (Sánchez-Rodríguez et al., 2020); the figure has been included under these lines to facilitate its visualization by the reviewer). This histological characterization further supports the robustness of our model and reinforces our confidence in using it to address new experimental questions, knowing that this foundational aspect has been thoroughly validated.

Supplementary file 1 in (Sánchez-Rodríguez et al., 2020)

2. It appears from figure 2 that learning differences between vehicle and disease animals narrows over time (e.g. Day 12) - do the authors feel effect on spatial memory is persistent?

We appreciate the reviewer's comments and acknowledge that the difference between groups appears to decrease over time. This may be because $\text{oA}\beta_{1-42}$ -treated mice do not experience a complete disruption of this type of memory but rather a significant reduction compared to control animals. As a result, their performance improves with repeated testing. However, their learning remains consistently lower than that of the control group, as confirmed by the post-hoc analysis of the two-way ANOVA (day 1: $p = 0.0145$; day 3: $p = 0.0147$; day 12: $p = 0.0276$).

To further confirm this effect, we conducted a Cox proportional hazards model, treating group as a categorical variable and number of days as a continuous variable. The analysis revealed significant effects of both treatment and time on success rates. Specifically, $\text{oA}\beta_{1-42}$ administration reduced the likelihood of reaching the escape hole by 72% (Hazard ratio = 0.28,

95% CI = 0.158–0.497, $p < 0.001$). Additionally, across all groups, the success rate improved by 26.3% per day (Hazard ratio = 1.263, 95% CI = 1.181–1.352, $p < 0.001$). (lines 480-490)

Therefore, we can confirm that the impairment in spatial memory is persistent. While oA β_{1-42} -treated mice do exhibit some learning over time, their performance remains significantly impaired compared to vehicle-treated mice. This supports our conclusion that a single injection of oA β_{1-42} is sufficient to induce a lasting decline in spatial memory.

3. It appears a broadband frequency increase in power was observed in the disease animals (Figure 3) which is surprising. Do the authors feel this reflects a spectral shift phenomenon? As I understand each LFP recording was normalised to pre-treatment which may introduce bias versus correcting across the mean/z score of all recordings.

Indeed, all the LFP recordings were normalized to the pre-treatment baseline. This decision was based on the fact that on the pre-treatment day, all animals were naïve, having not undergone any behavioral testing. Setting the pre-treatment LFPs as “100%” allowed us to assess changes on subsequent days—whether increases or decreases—attributable to learning and memory (in the vehicle group) or the presence of oA β_{1-42} .

This type of normalization has been widely used by our group, consistently yielding accurate results (Mayordomo-Cava *et al.*, 2020; Contreras *et al.*, 2023)), which gives us confidence that it does not introduce bias. Additionally, this approach enables us to track oscillatory activity changes within each group relative to their own pre-treatment baseline, providing a longitudinal perspective on treatment effects. Moreover, it maintains consistency with the other tests in the study (previous Figure 5; now Figure 7), facilitating a more intuitive interpretation of the data. This has been added to the Results section (page 19, lines 518-520).

The broadband frequency increase in power could be explained by compensatory mechanisms offsetting oA β 's initial hippocampal effects. Notably, we observed a reduction in theta and gamma spectral power in the hippocampus as soon as 1 h post-injection, whereas an increase in several rhythms' spectral power was noted in the PPC three days after amyloidosis induction. Despite utilizing separate cohorts to assess hippocampal and PPC oscillatory activities, the interconnectivity between these regions suggests the PPC may engage in compensatory actions to mitigate oA β 's adverse hippocampal impacts. Nevertheless, what our data conclusively show is the presence of aberrant oscillatory activity in both areas, which could underlie the memory alterations observed. This aligns with literature describing cortico-hippocampal circuit destabilization as a key early event in Alzheimer's disease progression (Palop & Mucke, 2010, 2016). Moreover, it is not ruled out that compensatory mechanisms are behind these alterations observed in these early stages of the disease (Styr & Slutsky, 2018).

4. Subsequent sliding window analysis (line 482) highlighted theta activity as most important factor in spectral change. This should be shown with appropriate statistics

We appreciate the observation made by the reviewer on the statistical report of the spectral analysis among the different rhythms (analysis intra-condition / inter-bands). Within each experimental condition (oA β_{1-42} and vehicle, both across the days: PRE, D-1, D-3, D-12), we conducted pairwise comparisons between *theta* vs. the other frequency bands (*delta*, *beta*, *low gamma*, and *high gamma*) using the Tukey-Kramer test for multiple comparisons. The results revealed that *theta* (4-12 Hz) rhythm centered at 8 Hz predominated over all other brain rhythms (see **new Figure 4A-B**):

Vehicle [Spectral power is reported as: mean \pm SD. **PRE:** *theta* (2.67 \pm 0.42) vs. *delta* (2.10 \pm 0.49), $p < 0.01$; *theta* vs. *beta* (1.06 \pm 0.13), $p < 0.01$; *theta* vs. low *gamma* (0.67 \pm 0.06), $p < 0.01$; *theta* vs. high *gamma* (0.37 \pm 0.04), $p < 0.01$. **D-1:** *theta* (2.11 \pm 0.56) vs. *delta* (1.79 \pm 0.75), $p < 0.01$; *theta* vs. *beta* (1.11 \pm 0.12), $p < 0.01$; *theta* vs. low *gamma* (0.75 \pm 0.06), $p < 0.01$; *theta* vs. high *gamma* (0.41 \pm 0.08), $p < 0.01$. **D-3:** *theta* (2.01 \pm 0.70) vs. *delta* (1.35 \pm 0.51), $p < 0.01$; *theta* vs. *beta* (1.09 \pm 0.16), $p < 0.01$; *theta* vs. low *gamma* (0.78 \pm 0.09), $p < 0.01$; *theta* vs. high *gamma* (0.44 \pm 0.09), $p < 0.01$. **D-12:** *theta* (2.60 \pm 0.30) vs. *delta* (2.21 \pm 0.54), $p < 0.01$; *theta* vs. *beta* (1.10 \pm 0.07), $p < 0.01$; *theta* vs. low *gamma* (0.68 \pm 0.06), $p < 0.01$; *theta* vs. high *gamma* (0.37 \pm 0.02), $p < 0.01$].

$\alpha\beta_{1-42}$ [Spectral power is reported as: mean \pm SD. **PRE:** *theta* (2.59 \pm 0.33) vs. *delta* (1.92 \pm 0.43), $p < 0.01$; *theta* vs. *beta* (1.11 \pm 0.13), $p < 0.01$; *theta* vs. low *gamma* (0.70 \pm 0.06), $p < 0.01$; *theta* vs. high *gamma* (0.37 \pm 0.04), $p < 0.01$. **D-1:** *theta* (2.27 \pm 0.55) vs. *delta* (1.52 \pm 0.74), $p < 0.01$; *theta* vs. *beta* (1.11 \pm 0.13), $p < 0.01$; *theta* vs. low *gamma* (0.78 \pm 0.08), $p < 0.01$; *theta* vs. high *gamma* (0.40 \pm 0.07), $p < 0.01$. **D-3:** *theta* (2.55 \pm 0.80) vs. *delta* (1.72 \pm 0.47), $p < 0.01$; *theta* vs. *beta* (1.17 \pm 0.20), $p < 0.01$; *theta* vs. low *gamma* (0.75 \pm 0.12), $p < 0.01$; *theta* vs. high *gamma* (0.36 \pm 0.09), $p < 0.01$. **D-12:** *theta* (3.88 \pm 0.85) vs. *delta* (2.72 \pm 0.60), $p < 0.01$; *theta* vs. *beta* (1.64 \pm 0.25), $p < 0.01$; *theta* vs. low *gamma* (1.08 \pm 0.15), $p < 0.01$; *theta* vs. high *gamma* (0.91 \pm 0.16), $p < 0.01$].

As can be seen here, the statistical report is too extensive to be included in the main text, so we only indicated once the p -value that satisfies all the multiple comparisons that demonstrate the predominance of the spectral component *theta* with respect to the rest of the oscillations (page 21 lines 571-572), aspect that is also evidenced in the spectrograms of **Figure 4B-C (now Figure 5B-C)**.

5. As I understand LFP recording was performed after task performance not during, this should be acknowledged as a limitation. In addition can the authors confirm animals were awake and alert during this period

We are grateful for the referee's feedback highlighting the need for clarification on our experimental design. LFP recordings were performed immediately after the behavioral task in D-1, D-3 and D-12, in a registration box in alert, freely moving mice, as the nature of the behavioral tests we conducted does not allow for simultaneous *in vivo* recordings. During PRE-treatment day, LFPs were recorded in that same box, and approximately at the same time of day to avoid influences of circadian rhythms. The necessity for animals to freely enter an escape hole in the Barnes maze task precludes the possibility of attaching wires to their heads, as this would inhibit their movement and ability to complete the task. Similarly, the OF task was performed in a the LABORAS[®] system, a platform that converts mechanical vibrations from animal movements into electrical signals. The sensitivity of this system means that any additional vibrations, such as those from a wire, could distort the results, undermining the integrity of the data collected. Also, LABORAS[®] system requires calibration before each session, including the animal's weight as a parameter. Therefore, introducing any external elements into the cage, such as wires, could interfere with the animals' behavior and compromise the accuracy of the measurements.

In summary, LFPs were recorded post-task, with animals fully awake and freely moving during the session.

This clarification has been made in the Method section of the ms. (page 9 line 227-228).

6. The authors found that theta and low gamma power were decreased within the hippocampus in the disease animals, was any theta-gamma coupling analysis attempted. In

addition, was repeated longer term assessment (e.g. days later) attempted as with the PPC experiments.

We thank the reviewer for raising this important point regarding both the timing of hippocampal assessments and the potential for theta–gamma coupling analysis. The lack of long-term analysis in the hippocampus is due to our decision to conduct a second cohort of animals focused on assessing hippocampal function in the short term after observing no effects in the PPC at early time points. Our aim was to analyze the temporal progression of the deleterious effects caused by $\text{oA}\beta_{1-42}$. Therefore, in this second cohort, we chose to examine early post-injection time points, starting at 1-hour post-injection, and included a behavioral test such as the OF, which also allows for the assessment of short-term learning. Consequently, we designed experiments in this second cohort specifically to investigate the role of the hippocampus shortly after $\text{oA}\beta_{1-42}$ injection.

Regarding the referee's query about the absence of *theta-gamma* coupling analysis, it was excluded from our study due to inconclusive results from such analyses. This decision was made to maintain the integrity of our findings and focus on areas where our data provided clear insights. However, we performed a comodulation analysis (cross-frequency coupling, see (Masimore *et al.*, 2004)) for the PPC's LFPs (see new Figure 6). This analysis revealed moderate-to-weak comodulations between low-frequency rhythms, notably *theta-beta* interactions (pre-injection and 12 days post-injection) and *delta-beta* interactions (3 days post-injection). In contrast, no significant *theta-gamma* interactions were detected.

7. Again relation between amount of Abeta expression within the hippocampus and LTP result (figure 5) would be important to understand

As we previously noted, in a recent study we showed that even 17 days after a single injection, although $\text{oA}\beta_{1-42}$ is no longer detectable in the hippocampus, its detrimental effects on LTP persist (Jiménez-Herrera *et al.*, 2023). This indicates that the observed alterations are not directly linked to the presence or concentration of $\text{oA}\beta_{1-42}$ in the hippocampus. Rather, the icv. injection of $\text{oA}\beta_{1-42}$ appears to initiate a cascade of events that leads to sustained synaptic dysfunction, as we have also shown in our recent studies (Djebari *et al.*, 2025; Mulero-Franco *et al.*, 2025), even in the absence of detectable peptide in this region.

- Contreras A, Djebari S, Temprano-Carazo S, Múnera A, Gruart A, Delgado-García JM, Jiménez-Díaz L & Navarro-López JD. (2023). Impairments in hippocampal oscillations accompany the loss of LTP induced by GIRK activity blockade. *Neuropharmacology* **238**, 109668.
- Djebari S, Jimenez-Herrera R, Iborra-Lazaro G, Jimenez-Díaz L & Navarro-Lopez JD. (2025). Social and contextual memory impairments induced by Amyloid-beta oligomers are rescued by Sigma-1 receptor activation. *Biomed Pharmacother* **184**, 117914.
- Jiménez-Herrera R, Contreras A, Djebari S, Mulero-Franco J, Iborra-Lázaro G, Jeremic D, Navarro-López J & Jiménez-Díaz L. (2023). Systematic characterization of a non-transgenic A β (1-42) amyloidosis model: synaptic plasticity and memory deficits in female and male mice. *Biol Sex Differ* **14**, 59.
- Masimore B, Kakalios J & Redish AD. (2004). Measuring fundamental frequencies in local field potentials. *J Neurosci Methods* **138**, 97-105.
- Mayordomo-Cava J, Iborra-Lázaro G, Djebari S, Temprano-Carazo S, Sánchez-Rodríguez I, Jeremic D, Gruart A, Delgado-García JM, Jiménez-Díaz L & Navarro-López JD. (2020). Impairments of Synaptic Plasticity Induction Threshold and Network Oscillatory Activity in the Hippocampus Underlie Memory Deficits in a Non-Transgenic Mouse Model of Amyloidosis. *Biology (Basel)* **9**.
- Mulero-Franco J, Jimenez-Herrera R, Contreras A, Djebari S, Jimenez-Díaz L & Navarro-Lopez JD. (2025). VU0810464, a selective GIRK channel activator, improves hippocampal-dependent synaptic plasticity and memory disrupted by amyloid-beta oligomers. *Biomed Pharmacother* **189**, 118247.
- Palop JJ & Mucke L. (2010). Amyloid-beta-induced neuronal dysfunction in Alzheimer's disease: from synapses toward neural networks. *Nat Neurosci* **13**, 812-818.
- Palop JJ & Mucke L. (2016). Network abnormalities and interneuron dysfunction in Alzheimer disease. *Nat Rev Neurosci* **17**, 777-792.
- Sánchez-Rodríguez I, Djebari S, Temprano-Carazo S, Vega-Avelaira D, Jiménez-Herrera R, Iborra-Lázaro G, Yajeya J, Jiménez-Díaz L & Navarro-López JD. (2020). Hippocampal long-term synaptic depression and memory deficits induced in early amyloidopathy are prevented by enhancing G-protein-gated inwardly rectifying potassium channel activity. *J Neurochem* **153**, 362-376.
- Styr B & Slutsky I. (2018). Imbalance between firing homeostasis and synaptic plasticity drives early-phase Alzheimer's disease. *Nat Neurosci* **21**, 463-473.

Dear Dr Navarro López,

Re: JP-RP-2025-286196R1 **"Posterior parietal cortex oscillatory activity reflects persistent spatial memory impairments induced by early hippocampal amyloidosis in male mice"** by Souhail Djebari, Ana Contreras, Victor D. Castro-Andres, Raquel Jimenez-Herrera, Guillermo Iborra-Lazaro, Raudel Sanchez-Campusano, Lydia Jimenez-Diaz, and Juan D. Navarro López

Thank you for submitting your manuscript to The Journal of Physiology. It has been assessed by a Reviewing Editor and by 3 expert referees and we are pleased to tell you that it is acceptable for publication following satisfactory revision.

REVISION CHECKLIST:

Please upload two versions of your manuscript text: one with all relevant changes highlighted and one clean version with no changes tracked. The manuscript file should include all tables and figure legends, but each figure/graph should be uploaded as separate, high-resolution files. The journal is now integrated with Wiley's Image Checking service. For further details,

see: <https://www.wiley.com/en-us/network/publishing/research-publishing/trending-stories/upholding-image-integrity-wileys-image-screening-service>

We look forward to receiving your revised submission.

Yours sincerely,

Nathan Schoppa
Senior Editor
The Journal of Physiology

REQUIRED ITEMS

- You must start the Methods section with a paragraph headed Ethical approval (https://jp.msubmit.net/cgi-bin/main.plex?form_type=display_requirements#methods).

Research must comply with The Journal's policies regarding animal experiments (<https://physoc.onlinelibrary.wiley.com/hub/animal-experiments>) and adherence to these policies must be stated in the manuscript.

Authors should confirm in their Methods section that their experiments were carried out according to the guidelines laid down by their institution's animal welfare committee, including an ethics approval reference number. The Methods section must contain a statement about access to food, water and housing, details of the anaesthetic regime: anaesthetic used, dose and route of administration, and method of killing the experimental animals.

- Papers must comply with the Statistics Policy: https://jp.msubmit.net/cgi-bin/main.plex?form_type=display_requirements#statistics.

In summary:

- If $n \leq 30$, all data points must be plotted in the figure in a way that reveals their range and distribution. A bar graph with data points overlaid, a box and whisker plot or a violin plot (preferably with data points included) are acceptable formats.
- If $n > 30$, then the entire raw dataset must be made available either as supporting information, or hosted on a not-for-profit repository, e.g. FigShare, with access details provided in the manuscript.
- 'n' clearly defined (e.g. x cells from y slices in z animals) in the Methods. Authors should be mindful of pseudoreplication.
- All relevant 'n' values must be clearly stated in the main text, figures and tables.
- The most appropriate summary statistic (e.g. mean or median and standard deviation) must be used. Standard Error of the Mean (SEM) alone is not permitted.
- Exact p values must be stated. Authors must not use 'greater than' or 'less than'. Exact p values must be stated to three significant figures even when 'no statistical significance' is claimed.

EDITOR COMMENTS

Reviewing Editor:

Ethics Concerns:

Please, indicate methods of euthanasia of animals, prior to immunohistochemical analysis.

Comments to the Author:

Thank you for the revised manuscript. Most of the reviewers' comments have been addressed; however, Reviewer 1 still has concerns regarding the statistical analysis of the data.

Senior Editor:

Comments for Authors to ensure the paper complies with the Statistics Policy:

N values are not consistently described in the figure legends. For example, what are likely to be N values are provided on the figure panels in Fig. 8. The authors should have a consistent convention.

Comments to the Author:

Your revised manuscript has been evaluated by the two reviewers of the original manuscript a Statistics Editor, and a reviewing editor, who appreciated the extensive changes that were made in the study. However, some major concerns in the area of statistical analyses remain. Reviewer 1 was not satisfied with how the authors responded to his/her prior concern around the authors' use of multiple measurements from the same animal. The study has also been evaluated by the Journal's statistics editor, who raises the same concern in the treatment of some of the data. As this is a concern that was brought up in the first review and not adequately addressed, the authors must adequately respond to these concerns in another revision, which will be re-evaluated. The Journal encourages the authors to seek input from a colleague with expertise in statistics. They should also re-review how they analyzed all of their data, not just the points raised by the reviewers.

The Journal's decision to allow submission of one additional revision is based on the generally positive response of the reviewers to the first submission, where it was pointed that the study is addressing an important question and the experiments well-done. Also, the authors did make extensive other changes in their latest revision. We cannot however guarantee acceptance of a next revision, if the conclusions of the study change with appropriate statistical analyses or if there are additional concerns in the analyses.

The authors should also respond to the reviewing editor's point about methods of euthanasia.

REFeree COMMENTS

Referee #1:

In my previous review I suggested that the authors have a statistician review their analyses because some df seemed incorrect. Now they clarify that they use in lines 580ff "the time-points from 1 to 299s" as independent measurements. This is blatantly wrong. Also, later (lines 536ff), 2 groups of 10 with 4 time points each does not give 1,72 df but 3,54 or 1,18 depending on analysis.

Figure 1 C still looks like LFPs were recorded during the task.

Referee #2:

The authors have mostly addressed points made in their revision. The spectral analysis element of the study is now clearer, as is methodology e.g. when LFP recordings were taken following behavioural testing. No further comments to add.

Referee #3 Statistics Editor:

Having reviewed both the referee's concerns and the authors' responses, I find that the revisions made to Figure 4 appropriately address the original issue. The reanalysis now treats the animal as the correct unit of replication, and the revised degrees of freedom are consistent with the experimental design. On this point, the authors' clarification and amended reporting satisfactorily resolve the referee's objection.

In contrast, the statistical approach underpinning Figure 5 remains problematic. The current analysis appears to treat trials or time bins as independent observations, resulting in highly inflated degrees of freedom and artificially small p values. While this approach may be suitable for generating descriptive estimates of spectral power, it is not valid for inferential comparisons between groups, as the animal remains the true experimental unit. Group-level inference would require either per-animal summaries or a mixed-effects framework that accounts for repeated measures. As such, I do not consider the

statistical claims reported in Figure 5 to be robust in their present form, and I would recommend that the authors either reanalyse these data accordingly or present the findings in descriptive terms with the appropriate caveats.

In addition, the statistical analysis section would benefit from greater clarity to ensure transparency and reproducibility. It is not stated how assumptions of normality and sphericity were assessed and handled. Given the repeated-measures design, it would be important to specify, for example, whether Mauchly's test or Greenhouse-Geisser epsilons were used and whether Greenhouse-Geisser corrections were applied when necessary.

END OF COMMENTS

POINT-BY-POINT REPLY TO THE REVIEWS

We sincerely appreciate the opportunity to revise and resubmit our manuscript. In response to the reviewers' recommendations, we have systematically re-evaluated all statistical analyses and clarified the corresponding methodologies throughout the manuscript. Additionally, we have revised our approach to the data presented in Figure 5, adopting a more descriptive analytical framework that better aligns with the nature of the data.

We have also made the necessary updates to the Methods section to ensure full compliance with the journal's policies and standards. All modifications have been clearly marked in red in the revised manuscript.

We believe that we have now addressed all the concerns raised by the reviewers, to whom we are sincerely grateful for their valuable feedback, and that the revised manuscript is now presented in a more rigorous and complete form.

EDITOR COMMENTS

Reviewing Editor:

Ethics Concerns: Please, indicate methods of euthanasia of animals, prior to immunohistochemical analysis.

We thank the editor for the observation. This important information has now been included y the Material and Methods section (page 8, lines 218-219).

Comments to the Author (Required):

Thank you for the revised manuscript. Most of the reviewers' comments have been addressed; however, Reviewer 1 still has concerns regarding the statistical analysis of the data.

Senior Editor:

Comments for Authors to ensure the paper complies with the Statistics Policy (Required): N values are not consistently described in the figure legends. For example, what are likely to be N values are provided on the figure panels in Fig. 8. The authors should have a consistent convention.

Information regarding the N values has now been added to all figure legends to ensure the ms. complies with the journal's Statistics Policy.

Comments to the Author:

Your revised manuscript has been evaluated by the two reviewers of the original manuscript and a reviewing editor, who appreciated the extensive changes that were made in the study. However, some major concerns in the area of statistical analyses remain. Reviewer 1 was not satisfied with how the authors responded to his/her prior concern around the authors' use of multiple measurements from the same animal. The study has also been evaluated by the Journal's statistics editor, who raises the same concern in the treatment of some the data. As this is a concern that was brought up in the first review and not adequately addressed, the authors must adequately respond to these concerns in another revision, which will be re-evaluated. The Journal encourages the authors to seek input from a colleague with expertise in statistics. They should also re-review how they analyzed all of their data, not just the points raised by the reviewers.

The Journal's decision to allow submission of one additional revision is based on the generally positive response of the reviewers to the first submission, where it was pointed that the study is addressing an important question and the experiments well-done. Also, the authors did make extensive other changes in their latest revision. We cannot however guarantee acceptance of a next revision, if the conclusions of the study change with appropriate statistical analyses or if there are additional concerns in the analyses.

The authors should also respond to the reviewing editor's point about methods of euthanasia.

REFEREE COMMENTS

Referee #1:

In my previous review I suggested that the authors have a statistician review their analyses because some df seemed incorrect. Now they clarify that they use in lines 580ff "the time-points from 1 to 299s" as independent measurements. This is blatantly wrong.

We greatly appreciate the referee's insightful comment, which led us to adopt a more appropriate statistical approach in this section of the Results (corresponding to Figure 5A and D). We acknowledge that, in the previous version of the ms., we "*over-analyzed*" the information resulting from the statistical inference matrices, considering the time steps as if they were independent measurements. This is inappropriate, without having demonstrated linear independence between successive time windows. Furthermore, this approach does not consider the number of experimental subjects as an independent measurement in the statistical report.

In the new version of the ms., we have adopted a descriptive approach using pie chart representations (**pages 11, lines 295-297**) to visualize the probability density percentages of the inferences [types ± 1 ($p < 0.05$) and type 0 ($p > 0.05$)] resulting from multiple comparisons between pairs of spectrograms in the *theta* frequency band (4-12 Hz). This new descriptive approach is also in line with the suggestion made by Reviewer #3. Accordingly, panels A and D of Figure 5 and its legend have been updated (**pages 24-25 of the new version of the ms.**), and the corresponding Results section has been rewritten to remove inconsistent statistical reporting and clarify the rationales for this change (**pages 22-23, lines 582-620**).

Regarding the mTFT method, two new key citations have been added: Jarvis *et al.*, 2001; Bokil *et al.*, 2007 (**page 10, lines 271-274**), which have also been included in the References section (**page 39, lines 1102-1103 and page 41, lines 1212-1213**).

Also, later (lines 536ff), 2 groups of 10 with 4 time points each does not give 1,72 df but 3,54 or 1,18 depending on analysis.

Regarding the referee's query about the degrees of freedom corresponding to the statistical reports for Figure 4, we would like to clarify this point. As we previously noted (see **page 20, lines 554-560 of the new version of the ms.**), two-way ANOVA F-test for the mean powers from both groups was applied. Degrees of freedom were obtained as: $df1 = \# \text{ of groups} - 1 = 1$ and $df2 = (\# \text{ of independent measurements} - 1) \times (\# \text{ of total experimental conditions}) = 72$. Here, the $\#$ of groups were equal to 2 (oA β_{1-42} and vehicle mice), the $\#$ of independent measurements were equal to 10 (number of mice in each group) and the $\#$ of total experimental conditions ($2 \times \#$ of days) was equal to 8. For a more detailed explanation of two-way ANOVA table, see the following screenshot from the MATLAB Help Center:

ANOVA takes the variation due to the factor or interaction and compares it to the variation due to error. If the ratio of the two variations is high, then the effect of the factor or the interaction effect is statistically significant. You can measure the statistical significance using a test statistic that has an F -distribution.

For the null hypothesis that the mean response for groups of the row factor A are equal, the test statistic is

$$F = \frac{SS_B / m - 1}{SSE / mk(R - 1)} \sim F_{m-1, mk(R-1)}.$$

For the null hypothesis that the mean response for groups of the column factor B are equal, the test statistic is

$$F = \frac{SS_A / k - 1}{SSE / mk(R - 1)} \sim F_{k-1, mk(R-1)}.$$

For the null hypothesis that the interaction of the column and row factors are equal to zero, the test statistic is

$$F = \frac{SS_{AB} / (m - 1)(k - 1)}{SSE / mk(R - 1)} \sim F_{(m-1)(k-1), mk(R-1)}.$$

If the p -value for the F -statistic is smaller than the significance level, then ANOVA rejects the null hypothesis. The most common significance levels are 0.01 and 0.05.

ANOVA Table

The ANOVA table captures the variability in the model by the source, the F -statistic for testing the significance of this variability, and the p -value for deciding on the significance of this variability. The p -value returned by `anova2` depends on assumptions about the random disturbances, ϵ_{ij} , in the model equation. For the p -value to be correct, these disturbances need to be independent, normally distributed, and have constant variance. The standard ANOVA table has this form:

Source	Sum of Squares (SS)	Degrees of Freedom	Mean Squares (MS)	F -statistic	p -value
Columns	SS_A	$k - 1$	MS_A	MS_A / MSE	$P(F_{k-1, mk(R-1)} > F)$
Rows	SS_B	$m - 1$	MS_B	MS_B / MSE	$P(F_{m-1, mk(R-1)} > F)$
Interaction	SS_{AB}	$(m - 1)(k - 1)$	MS_{AB}	MS_{AB} / MSE	$P(F_{(m-1)(k-1), mk(R-1)} > F)$
Error	SSE	$mk(R - 1)$	MSE		
Total	SST	$mkR - 1$			

Here $m = \#$ of animal groups = 2 (oA β_{1-42} and vehicle mice), $k = \#$ of days = 4 (PRE, D-1, D-3, D-12), and $R = \#$ of independent measurements = 10 mice. We used F -statistics to do null hypotheses tests to find out if the spectral power is the same across days (PRE, D-1, D-3, D-12), animal groups (oA β_{1-42} and vehicle mice), and day - group pairs. The days and animal groups appear to have no interaction. As the interaction was not significant, the main effects (for days or animal groups) were interpreted independently. Therefore, Fisher's statistic test was reported as $F_{(df1, df2)}$ with $df1 = m - 1 = 1$ and $df2 = mk(R - 1) = 72$.

For example, the statistical report for the animal group effect (rows factor) was $F_{(1,72)} = 5.70$, $p = 0.0197$ for θ band, which was highly significant. This p -value indicates that one mice group was outperforming the other in the mean spectral power of θ band. The observed p -value indicates that an F -statistic as extreme as the observed F occurs by chance about twenty (19.7) out of 1000 times, if the spectral powers of θ band were truly equal from group to group (see the following report for θ band):

ANOVA Table					
Source	SS	df	MS	F	Prob>F
Columns	2797.8	3	932.6	2.2	0.0957
Rows	2413.7	1	2413.74	5.69	0.0197 *
Interaction	2424.7	3	808.23	1.9	0.1365
Error	30550.9	72	424.32		
Total	38187.2	79			

Figure 1 C still looks like LFPs were recorded during the task.

We agree with the reviewer that the figure could be misleading. Therefore, we have made a minor adjustment to improve clarity and have explained in the figure legend (pages 7-8, lines 182-183 and 187).

Referee #2:

The authors have mostly addressed points made in their revision. The spectral analysis element of the study is now clearer, as is methodology e.g. when LFP recordings were taken following behavioural testing. No further comments to add.

We thank Reviewer #2 for the positive evaluation of the revisions made to the manuscript.

Referee #3:

Comments for Author (Required):

Having reviewed both the referee's concerns and the authors' responses, I find that the revisions made to Figure 4 appropriately address the original issue. The reanalysis now treats the animal as the correct unit of replication, and the revised degrees of freedom are consistent with the experimental design. On this point, the authors' clarification and amended reporting satisfactorily resolve the referee's objection.

We thank Reviewer #3 for positively evaluating the revisions made to Figure 4.

In contrast, the statistical approach underpinning Figure 5 remains problematic. The current analysis appears to treat trials or time bins as independent observations, resulting in highly inflated degrees of freedom and artificially small p values. While this approach may be suitable for generating descriptive estimates of spectral power, it is not valid for inferential comparisons between groups, as the animal remains the true experimental unit. Group-level inference would require either per-animal summaries or a mixed-effects framework that accounts for repeated measures. As such, I do not consider the statistical claims reported in Figure 5 to be robust in their present form, and I would recommend that the authors either reanalyse these data accordingly or present the findings in descriptive terms with the appropriate caveats.

We are grateful for the referee's feedback highlighting the need for clarification on our statistical approach corresponding to Figure 5. We acknowledge that, in the previous version of the ms., we "*over-analyzed*" the information resulting from the statistical inference matrices, considering the time steps as if they were independent measurements. This is inappropriate, without having demonstrated linear independence between successive time windows. Furthermore, this approach does not consider the number of experimental subjects as an independent measurement in the statistical report.

In the new version of the manuscript, and following the referee's recommendations, we have adopted a descriptive approach using pie charts to visualize the probability density percentages of the inferences [types ± 1 ($p < 0.05$) and type 0 ($p > 0.05$)] obtained from multiple comparisons between pairs of spectrograms in the theta frequency band (4–12 Hz). This new approach, suggested by the present referee and adopted in the revised version, has also helped us to address specific methodological points raised by Referee #1. Accordingly, panels A and D of Figure 5 have been updated (**page 24 of the new version of the ms.**), and the corresponding Results section has been rewritten to remove inconsistent statistical reporting and clarify the rationales for this change (**pages 22-23, lines 582-620**).

In addition, the statistical analysis section would benefit from greater clarity to ensure transparency and reproducibility. It is not stated how assumptions of normality and sphericity were assessed and handled. Given the repeated-measures design, it would be important to specify, for example, whether Mauchly's test or Greenhouse-Geisser epsilons were used and whether Greenhouse-Geisser corrections were applied when necessary.

We thank the referee for his/her observation and fully agree on the importance of this information. Accordingly, the requested details have been added to the Materials and Methods section (**page 17, lines 466-467 and 469-470**).

Re: JP-RP-2025-286196R2 **"Posterior parietal cortex oscillatory activity reflects persistent spatial memory impairments induced by early hippocampal amyloidosis in male mice"** by Souhail Djebari, Ana Contreras, Victor D. Castro-Andres, Raquel Jimenez-Herrera, Guillermo Iborra-Lazaro, Raudel Sanchez-Campusano, Lydia Jimenez-Diaz, and Juan D. Navarro López

Dear Dr Navarro López,

Thank you for submitting your manuscript to The Journal of Physiology. It has been assessed by a Reviewing Editor and by 3 expert referees and we are pleased to tell you that it is potentially acceptable for publication following satisfactory major revision.

Please address all the points raised and incorporate all requested revisions or explain in your Response to Referees why a change has not been made. We hope you will find the comments helpful and that you will be able to return your revised manuscript within 2 months. If your article is NOT for a Special Issue, you may have 9 months to revise. If you require an extension, please contact journal staff: jp@physoc.org. Please note that this letter does not constitute a guarantee for acceptance of your revised manuscript.

LANGUAGE EDITING AND SUPPORT FOR PUBLICATION: If you would like help with English language editing, or other article preparation support, Wiley Editing Services offers expert help, including English Language Editing, as well as translation, manuscript formatting, and figure formatting at www.wileyauthors.com/eoo/preparation. You can also find resources for Preparing Your Article for general guidance about writing and preparing your manuscript at www.wileyauthors.com/eoo/prepresources.

REVISION CHECKLIST:

Upload a full Response to Referees file. To create your 'Response to Referees': copy all the reports, including any comments from the Senior and Reviewing Editors, into a Microsoft Word, or similar, file and respond to each point, using

font or background colour to distinguish comments and responses and upload as the required file type.

We look forward to receiving your revised submission.

Yours sincerely,

Nathan Schoppa
Senior Editor
The Journal of Physiology

REQUIRED ITEMS FOR REVISION:

- You must start the Methods section with a paragraph headed Ethical approval (https://jp.msubmit.net/cgi-bin/main.plex?form_type=display_requirements#methods).

Research must comply with The Journal's policies regarding animal experiments (<https://physoc.onlinelibrary.wiley.com/hub/animal-experiments>) and adherence to these policies must be stated in the manuscript.

Authors should confirm in their Methods section that their experiments were carried out according to the guidelines laid down by their institution's animal welfare committee, including an ethics approval reference number. The Methods section must contain a statement about access to food, water and housing, details of the anaesthetic regime: anaesthetic used, dose and route of administration, and method of killing the experimental animals.

- Papers must comply with the Statistics Policy: https://jp.msubmit.net/cgi-bin/main.plex?form_type=display_requirements#statistics.

In summary:

- If $n \leq 30$, all data points must be plotted in the figure in a way that reveals their range and distribution. A bar graph with data points overlaid, a box and whisker plot or a violin plot (preferably with data points included) are acceptable formats.
- If $n > 30$, then the entire raw dataset must be made available either as supporting information, or hosted on a not-for-profit repository, e.g. FigShare, with access details provided in the manuscript.
- 'n' clearly defined (e.g. x cells from y slices in z animals) in the Methods. Authors should be mindful of pseudoreplication.
- All relevant 'n' values must be clearly stated in the main text, figures and tables.
- The most appropriate summary statistic (e.g. mean or median and standard deviation) must be used. Standard Error of the Mean (SEM) alone is not permitted.
- Exact p values must be stated. Authors must not use 'greater than' or 'less than'. Exact p values must be stated to three significant figures even when 'no statistical significance' is claimed.

EDITOR COMMENTS

Reviewing Editor:

Many thanks for revising the manuscript and addressing most of the raised issues. There is still concern about the statistics used on the data - given that the same group of animals was tested across multiple time points, a repeated measurement two-way ANOVA should be utilized to estimate significance between the groups.

Senior Editor:

The last revision of your manuscript was reviewed by the two prior external referees, JP's statistics editor, and the Reviewing Editor (RE), and, while some of the prior points raised about the statistical analyses were well-addressed, other concerns around statistics remain. Some of the remaining concerns were fleshed out in further conversations between myself, the RE, and the statistics editor. The remaining concerns, listed below, will need to be addressed in another revision of the manuscript that will be reviewed by JP's statistics editor, the RE, and me.

The remaining concerns include:

(1) Most important is that there remains lack of clarity of whether the proper two-way ANOVA test was used in several instances in the study (including around Fig. 3B, Fig. 4D,H, and Fig 7D,H and J,K). In each of those cases, the likely appropriate test is a repeated measures two-way ANOVA or a two-way mixed ANOVA, since the same animals were tested across multiple days. Your Methods section indicates that you generally used both two-way and repeated measures two-way ANOVAs, but, in the Results section where individual experiments and analyses are described, the tests are presented as two-way ANOVAs. Please check that the correct tests were used and, at a bear minimum, provide clarification of exactly what test was used in each instance and the rationale for the use of the test that can be clearly understood by readers.

(2) Around Fig. 4D,E, results are described in the main text as if there was an effect of time, but significance in the statistical analysis described is reported only for treatment group, not time. This inconsistency needs to be corrected.

(3) Minor point: the placement of the significance asterisks with horizontal lines in several places in Fig. 4 and in Fig. 7G makes it seem as if significance is being reported for comparisons between the days, when in fact the comparisons are between treatment groups. This needs to be corrected. Asterisks should be assigned to single days.

REFEREE COMMENTS

Referee #1:

There were two statistical problems (that were pointed out).

The first one was solved by not presenting statistics anymore (presenting simple percentages instead). Not using statistics because the appropriate test is not significant is not a solution I approve of.

For the second problem, I pointed out that a test where 2 groups of 10 animals are tested 4 times each, should not have $df=1,72$ but either 1,18 (for the between-subject effects) or 3,54 (for the within-subject effects). It was suggested to the authors to get statistical help, but they preferred not to. Instead they refer to the Matlab manual. They still insist that a test with $df=1,72$ is the correct one. This shows that they have not used a repeated measures test, which would be appropriate in this case, but a test for 80 independent measures, which is wrong.

If they prefer not to ask a statistical expert, I can now also recommend ChatGPT, which has become very good at explaining statistics. Simply ask: "What's the degrees of freedom for a test where 2 groups of 10 animals are tested 4 times each, if I want to compare the two groups?" to get started.

Referee #2:

Thank you - my comments were previously addressed, nothing further

Referee #3:

This is an acceptable revision. The authors have demonstrated understanding of the statistical issue, adopted a more appropriate approach given their sample size, and importantly removed claims of robust statistical group differences. The approach isn't perfect (the underlying autocorrelation problem remains), but it's honest about the limitations and doesn't overstate the evidence.

END OF COMMENTS

Response to Editors

EDITOR COMMENTS

Reviewing Editor:

Many thanks for revising the manuscript and addressing most of the raised issues. There is still concern about the statistics used on the data - given that the same group of animals was tested across multiple time points, a repeated measurement two-way ANOVA should be utilized to estimate significance between the groups.

We thank the Reviewing Editor for the valuable feedback. In response to the concerns raised regarding the statistical analysis, and following the Reviewing Editor's recommendations as well as further advice, we have reanalyzed all the points that have generated controversy using a two-way repeated measures ANOVA. Statistical analyses are now correctly reported, clearly indicating the tests used. The results remain consistent with our original findings, and we believe they do not alter the significance of our work.

Senior Editor:

The last revision of your manuscript was reviewed by the two prior external referees, JP's statistics editor, and the Reviewing Editor (RE), and, while some of the prior points raised about the statistical analyses were well-addressed, other concerns around statistics remain. Some of the remaining concerns were fleshed out in further conversations between myself, the RE, and the statistics editor. The remaining concerns, listed below, will need to be addressed in another revision of the manuscript that will be reviewed by JP's statistics editor, the RE, and me.

We thank the Senior Editor for the feedback received and for the opportunity to revise our work to address any concerns related to the statistical analyses employed. Below, we respond to all the points raised, which we have carefully addressed in line with the recommendations provided, with the aim of preventing any further confusion.

The remaining concerns include:

(1) Most important is that there remains lack of clarity of whether the proper two-way ANOVA test was used in several instances in the study (including around Fig. 3B, Fig. 4D,H, and Fig 7D,H and J,K). In each of those cases, the likely appropriate test is a repeated measures two-way ANOVA or a two-way mixed ANOVA, since the same animals were tested across multiple days. Your Methods section indicates that you generally used both two-way and repeated measures two-way ANOVAs, but, in the Results section where individual experiments and analyses are described, the tests are presented as two-way ANOVAs. Please check that the correct tests were used and, at a bear minimum, provide clarification of exactly what test was used in each instance and the rationale for the use of the test that can be clearly understood by readers.

We would like to thank the Senior Editor for the detailed explanation, which helped us to fully understand that the statistical approach initially applied was not appropriate for

the repeated-measures nature of our data. Therefore, and following your recommendations as well as further advice, we have reanalyzed all relevant datasets using a two-way repeated measures ANOVA.

Accordingly, we have revised the Materials and Methods section to accurately reflect the statistical tests used (page 17, lines 456-466) and updated the Results section with the corrected statistical values (page 20, lines 536-537 and 543-544 and page 27, lines 714- 715). We have also specified the tests used in sections where this information was previously missing (lines 484-485, 535-536, 542, 684, 694, 713, 729, 780-781, 784-785). We believe these revisions improve clarity and consistency, yielding degrees of freedom similar to those noted by Referee 1 and helping to prevent further misunderstanding.

Importantly, the new statistical analyses continue to show significant treatment effects in all the frequency bands where they were previously reported and thus do not alter the overall significance of our findings. Minor changes were detected in the significance levels of the post-hoc analysis for Figure 4D–H; therefore, the Discussion section has been updated accordingly (page 32, lines 876-878).

(2) Around Fig. 4D,E, results are described in the main text as if there was an effect of time, but significance in the statistical analysis described is reported only for treatment group, not time. This inconsistency needs to be corrected.

We understand that the text may have been misleading, as the first part was intended solely to describe the graphical representation of the results across different time points, whereas the statistical analysis referred specifically to the treatment effect. In the revised version of the manuscript, we have rephrased this section to prevent any potential confusion (page 20, lines 529, 538-541).

(3) Minor point: the placement of the significance asterisks with horizontal lines in several places in Fig. 4 and in Fig. 7G makes it seem as if significance is being reported for comparisons between the days, when in fact the comparisons are between treatment groups. This needs to be corrected. Asterisks should be assigned to single days.

We agree with this observation. The significance asterisks were initially placed based on the statistical analysis conducted across both days combined. However, following the updated statistical analysis, we have revised Figures 4 and 7 so that the asterisks now reflect the results of the post-hoc comparisons for each day individually.

REFEREE COMMENTS

Referee #1:

There were two statistical problems (that were pointed out).

The first one was solved by not presenting statistics anymore (presenting simple percentages instead). Not using statistics because the appropriate test is not significant is not a solution I approve of.

For the second problem, I pointed out that a test where 2 groups of 10 animals are

tested 4 times each, should not have $df=1,72$ but either 1,18 (for the between-subject effects) or 3,54 (for the within-subject effects). It was suggested to the authors to get statistical help, but they preferred not to. Instead they refer to the Matlab manual. They still insist that a test with $df=1,72$ is the correct one. This shows that they have not used a repeated measures test, which would be appropriate in this case, but a test for 80 independent measures, which is wrong.

If they prefer not to ask a statistical expert, I can now also recommend ChatGPT, which has become very good at explaining statistics. Simply ask: "What's the degrees of freedom for a test where 2 groups of 10 animals are tested 4 times each, if I want to compare the two groups?" to get started.

We thank the reviewer for the comments on the statistical analyses. Following further consideration of the experimental design and taking into account the suggestions provided by both the reviewer and the editors, we have re-evaluated the statistical approach used for the datasets in question. All relevant datasets that were questioned have now been re-analyzed using a two-way repeated measures ANOVA, and statistical results, including non-significant findings, are now fully reported. The Methods, Results and figure legends have been revised accordingly.

Referee #2:

Thank you - my comments were previously addressed, nothing further

Referee #3:

This is an acceptable revision. The authors have demonstrated understanding of the statistical issue, adopted a more appropriate approach given their sample size, and importantly removed claims of robust statistical group differences. The approach isn't perfect (the underlying autocorrelation problem remains), but it's honest about the limitations and doesn't overstate the evidence.

Dear Dr Navarro López,

Re: JP-RP-2026-286196R3 "**Posterior parietal cortex oscillatory activity reflects persistent spatial memory impairments induced by early hippocampal amyloidosis in male mice**" by Souhail Djebari, Ana Contreras, Victor D. Castro-Andres, Raquel Jimenez-Herrera, Guillermo Iborra-Lazaro, Raudel Sanchez-Campusano, Lydia Jimenez-Diaz, and Juan D. Navarro López

We are pleased to tell you that your paper has been accepted for publication in The Journal of Physiology.

Yours sincerely,

Nathan Schoppa
Senior Editor
The Journal of Physiology

IMPORTANT POINTS TO NOTE FOLLOWING ACCEPTANCE OF YOUR PAPER:

- **IMPORTANT NOTICE ABOUT OPEN ACCESS:** To assist authors whose funding agencies mandate immediate public access to published research findings, The Journal of Physiology allows authors to pay an Open Access (OA) fee to have their papers made freely available immediately on publication.

- You can help your research get the attention it deserves! Check out Wiley's free Promotion Guide for best-practice recommendations for promoting your work at: www.wileyauthors.com/eoo/guide. You can learn more about Wiley Editing Services which offers professional video, design, and writing services to create shareable video abstracts, infographics, conference posters, lay summaries, and research news stories for your research at: www.wileyauthors.com/eoo/promotion.

- If you would like to receive our 'Research Roundup', a monthly newsletter highlighting the cutting-edge research published in The Physiological Society's family of journals (The Journal of Physiology, Experimental Physiology, Physiological Reports, The Journal of Nutritional Physiology and The Journal of Precision Medicine: Health and Disease), please click this link, fill in your name and email address and select 'Research Roundup': <https://www.physoc.org/journals-and-media/membernews>

EDITOR COMMENTS

Reviewing Editor:

Many thanks for a thorough revision of the manuscript - I have no further comments related to the study.

Senior Editor:

Thank you for sending the most recent revision of your manuscript to The Journal of Physiology. The work has been examined by the two original external referees, JP's statistics editor, and the reviewing editor, and the work is now considered to be acceptable for publication. We especially appreciate your additional effort at addressing the remaining concerns around statistics. Congratulations for a nice study!

REFeree COMMENTS

Referee #1:

I am now satisfied with the responses of the authors. I also thank the editors for their effort in clarifying the statistical inconsistencies.

Referee #2:

My comments were previously addressed in the first revision, nothing further to add

Referee #3:

In my view the statistical revisions are acceptable. The authors have demonstrated understanding of the repeated measures design requirements and have implemented appropriate corrections.